# Bidirectional Recurrence for Cardiac Motion Tracking with Gaussian Process Latent Coding

**Jiewen Yang    Yiqun Lin    Bin Pu    Xiaomeng Li**✉
The Hong Kong University of Science and Technology
{jyangcu, ylindw}@connect.ust.hk  {eebinpu, eexmli}@ust.hk

## Abstract

Quantitative analysis of cardiac motion is crucial for assessing cardiac function. This analysis typically uses imaging modalities such as MRI and Echocardiograms that capture detailed image sequences throughout the heartbeat cycle. Previous methods predominantly focused on the analysis of image pairs lacking consideration of the motion dynamics and spatial variability. Consequently, these methods often overlook the long-term relationships and regional motion characteristic of cardiac. To overcome these limitations, we introduce the **GPTrack**, a novel unsupervised framework crafted to fully explore the temporal and spatial dynamics of cardiac motion. The GPTrack enhances motion tracking by employing the sequential Gaussian Process in the latent space and encoding statistics by spatial information at each time stamp, which robustly promotes temporal consistency and spatial variability of cardiac dynamics. Also, we innovatively aggregate sequential information in a bidirectional recursive manner, mimicking the behavior of diffeomorphic registration to better capture consistent long-term relationships of motions across cardiac regions such as the ventricles and atria. Our GPTrack significantly improves the precision of motion tracking in both 3D and 4D medical images while maintaining computational efficiency. The code is available at: https://github.com/xmed-lab/GPTrack.

## 1 Introduction

Cardiac motion tracking from Cardiac Magnetic Resonance Imaging (MRI) and Echocardiograms is crucial in quantitative cardiac image processing. These imaging techniques provide comprehensive image sequences that cover an entire heartbeat cycle, allowing for detailed analysis of cardiac dynamics. Conventional non-parametric cardiac motion tracking approaches, such as B-splines [1], Demons algorithms [2] and optical-flow based methods [3, 4, 5], are commonly utilized due to their flexibility and ability to align detailed structures within images. However, these methods face significant challenges in motion tracking because they lack topology-preserving constraints and temporal coherence. The diffeomorphic registration method [6, 7, 8], which formulates the registration process as a group of diffeomorphisms in Lagrangian dynamics, is a nice candidate for topology-preserving motion tracking. However, traditional optimization-based diffeomorphic registration methods are computationally intensive and sensitive to noises, hindering their applications in efficient cardiac motion tracking.

Current deep learning-based techniques [9, 10, 11, 12, 13, 14, 15, 16] employ these advanced imaging modalities to register image pairs within the same patient, and some [14, 15, 16] adopt the diffeomorphic routine and learn the Lagrangian strain to describe the motion relationship between the reference frame and subsequent frames, aggregating dynamic information into consecutive cardiac motions as Lagrangian displacements. Although the above diffeomorphic methods better model the dynamic and continuous nature of cardiac motion, they still have room for improvement in handling the long-term temporal relationship in videos. For example, the approach [14] requires segmentation

38th Conference on Neural Information Processing Systems (NeurIPS 2024).

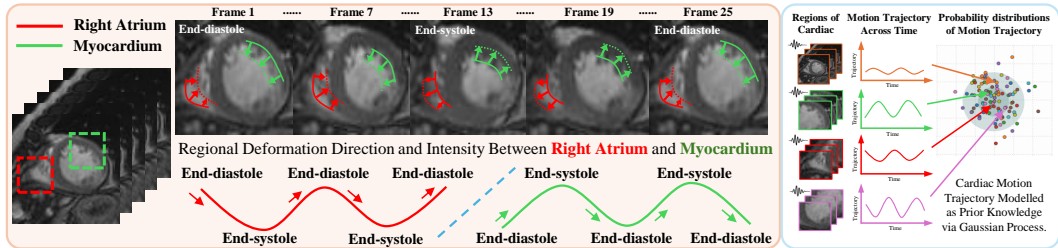

Figure 1: **Regional Motions in Cardiac:** The left sequential MRI frames within a heartbeat cycle illustrate that motion direction and intensity are completely different between the right atrium and myocardium during End-diastole and End-systole. **Formulate Cardiac Motion as Prior Knowledge:** The right figure depicts the regions of motion trajectory across the heartbeat cycle, alongside the probability distributions of motion trajectory. Curves (Middle) are the motion trajectory changes of different MRI sequences (Left). Highlighting the cardiac motion trajectory that follows a certain pattern can be modelled as prior knowledge via the Gaussian Process (Right).

annotation to generate dense motion trajectories and calculate Lagrangian strain. Additionally, the methods outlined in [15, 16] are prone to error accumulation as Lagrangian displacements are integrated without regularizing temporal variations. In spatial views, while the global trajectory flow may follow a specific pattern, significant variations exist within each region regarding the phases, amplitudes, and intensities of the motion. For example, Figure 1 shows regions of the Right Atrium (red) and Myocardium (green) performing the opposite trajectories during the heartbeat cycle. Conversely, similar regions across different cases exhibit consistent motions. Hence, ignoring the regional scale may lead to a fragmented understanding of cardiac motion, underscoring the need for more nuanced analytical approaches. Furthermore, as shown in the right of Figure 1, the deformation is bounded in the space of periodically specific human cardiac motion variation.

To leverage discussed temporal and spatial information for cardiac motion tracking, we propose a novel framework named **GPTrack**. Our GPTrack has several appealing facets: **1)** GPTrack employs the Gaussian Process (GP) to formulate the consistent temporal patterns in the latent space of diffeomorphic frameworks, promoting the consistency of cardiac motion; **2)** GPTrack utilizes position information in the latent space to encode the statistics of the sequential Gaussian process, by which we model the region-specific motion and obtain a more precise estimation related to cardiac motion; **3)** GPTrack leverages the inherent temporal continuity in cardiac motion by aggregating long-term relationships through forward and backward video flows, which mimics the forward-backward manner of classical diffeomorphic registration framework [6]. To evaluate the performance of GPTrack in cardiac motion tracking, we conduct experiments based on 3D Echocardiogram videos [17, 18] and 4D temporal MRI image [19]. Results in Tables 1, 2 and 3, show the GPTrack enhance the accuracy of motion tracking performance in a clear margin, without substantially increasing the computational cost in comparison to other state-of-the-art methods. **Our contributions are summarized as follows:**

**1.** We propose a novel cardiac motion tracking framework named the **GPTrack**. This framework employs the *Gaussian Process* (GP) to promote temporal consistency and regional variability in compact latent space, establishing a robust regularizer to enhance cardiac motion tracking accuracy.

**2.** The GPTrack framework is designed to capture the long-term relationship of cardiac motion via a bidirectional recursive manner, and its forward-backward manner mimics the workflows of the classical diffeomorphic registration framework. By this approach, our method provides a more accurate and reliable estimation of cardiac motion.

**3.** Our GPTrack framework achieves state-of-the-art performance on both 3D Echocardiogram videos and 4D temporal MRI datasets, maintaining comparable computational efficiency. The results demonstrate that our method adapts effectively across different medical imaging modalities, proving its utility in different clinical settings.

## 2 Related Work

### 2.1 Cardiac Motion Tracking via Non-parametric Registration Approach

Extensive works have been proposed to address registration by optimising within the space of displacement vector fields. Models related to elastic matching were proposed by [20, 21]. [22] utilized

statistical parametric mapping for improvement. Techniques incorporating free-form deformations with B-splines and Maxwell demons were adapted by [1] and [2], respectively. The Harmonic phase-based method, which utilizes spectral peaks in the Fourier domain for cardiac motion tracking, was introduced by [23]. This method calculates phase images from the inverse Fourier transforms and is specifically exploited in the analysis of tagged MRI. Popular formulations [6, 8, 24, 7] introduce topology-preserved diffeomorphic transforms. In the realm of diffeomorphic registration, inverse consistent diffeomorphic deformations have been estimated by [6] and [7]. Syn [24] proposed standard symmetric normalization. RDMM [8] considered regional parameterization based on Large Deformation Diffeomorphic Metric Mapping [6]. Despite their remarkable success in computational anatomy studies, these approaches are also time-consuming and susceptible to noise.

## 2.2 Cardiac Motion Tracking with Deep-Learning based Registration Method

Recent advancements in medical image registration have increasingly leveraged Deep Learning technologies. Pioneering studies [14, 25, 26, 27, 28] utilize ground truth displacement fields obtained by simulating deformations and deformed images, typically estimated using non-parametric methods. These approaches, however, may be limited by the types of deformations they can effectively model, which can affect both the quality and accuracy of the registration. Unsupervised methods, as discussed by [9, 10, 15, 29, 30, 31, 32] have shown promise by learning deformation through the warping of a fixed image to a moving image using spatial transformation functions [33]. These methods have been extended to include deformable models for single directional deformation field tracking [9, 26, 34] and diffeomorphic models for stationary velocity fields [32, 35, 36]. Further application of diffeomorphic models to cardiac motion tracking has been explored by [13, 15, 16, 30, 31]. These models predict motion fields that are both differentiable and invertible, ensuring one-to-one mappings and topology preservation. Recent studies in denoising diffusion probabilistic models (DDPM), such as [11] and [12], have achieved considerable success in registration tasks. However, DDPM-based methods face challenges in building temporal connections and demand substantial computational resources. The DL-based optical flow (OF) methods [37, 38, 39, 40] apply widely in nature image motion tracking. However, as illustrated in [7, 15, 16], due to annotations requirements and photometric constraints, they cannot be adopted in unsupervised cardiac motion tracking in medical image domains *(See Section A1 in Appendix for detailed discussion)*.

## 3 Methodology

### 3.1 Diffeomorphic Tracking of Cardiac Motion

Diffeomorphic motion tracking and registration techniques are widely used in medical image analysis because they seek topology-preserving mapping between source and target images [6, 8]. Formally, given the source image $x_0$ and target image $x_1$, the diffeomorphic registration aims at a family of differentiable and invertible mappings $\{\phi_t\}_{t \in [0,1]}$ with the boundary condition $\phi_0 = \mathrm{Id}$ and $\phi_1(x_0) = x_1$, where Id is the identity mapping. The diffeomorphism

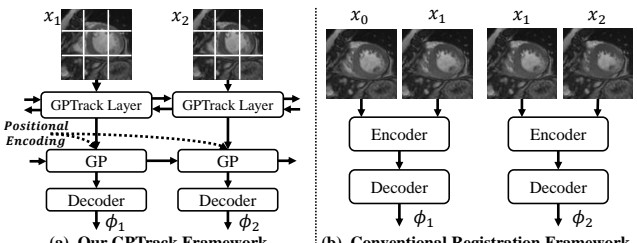

Figure 2: Comparsion between our GPTrack (a) and conventional registration framework (b).

$\phi_t$ can be parameterized as its derivatives (velocity field) $\mathbf{v}_t$ as follows:

$$\frac{d\phi_t}{dt} = \mathbf{v}_t(\boldsymbol{\phi}_t) := \mathbf{v}_t \circ \boldsymbol{\phi}_t \Longleftrightarrow \boldsymbol{\phi}_t = \phi_0 + \int_0^t \mathbf{v}_s(\boldsymbol{\phi}_s)ds, \ \ s \in [0,1], \tag{1}$$

where $\circ$ is the composition operator. For numerical implementation, the associative property of the diffeomorphism group indicates $\phi_{t_1+t_2} = \phi_{t_1} \circ \phi_{t_2}$, and the integral of Equation 1 can be approximated by $\phi_{t+\delta} \approx \phi_t + \mathbf{v}_t \delta$ for $t \in [0,1)$ and small enough $\delta$. In this study, we follow the parameter settings of [7, 15, 16, 35, 36] and take $\delta = \frac{1}{2^N}$, $N = 7$ to discretize the path of diffeomorphic deformation.

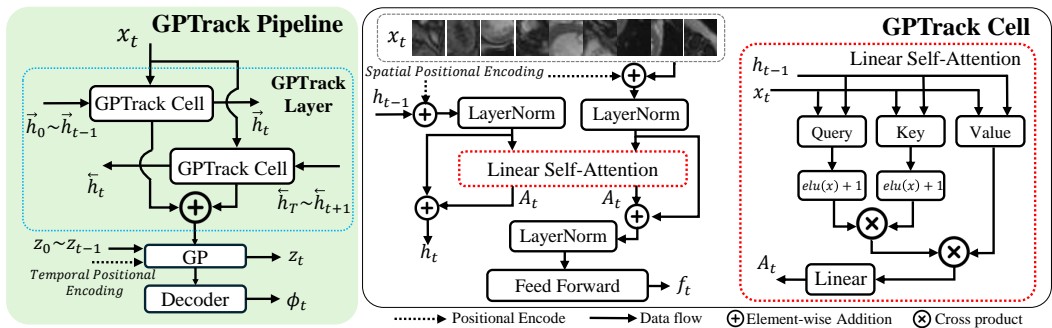

Figure 3: The overview pipeline of GPTrack (one layer). The $x$, $h$, $\dot{h}$ and $z$ denote input, forward hidden states, backward hidden states and latent coordinates. Feature $\vec{f}_t$ with probabilistic prior on the latent space via Gaussian Process then enters the decoder to predict the motion field $\phi$. Subscript $t$ denotes the $t$-th position in total $T$ moments. $elu(\cdot)$ represent the exponential linear units [42].

## 3.2 Motion Tracking with Gaussian Process Latent Coding

Our proposed method adopts the generative variational autoencoder (VAE) framework as the backbone of a diffeomorphic tracking network, as suggested by previous methods [15, 16, 36]. However, as shown in Figure 2, the first difference between our method and other methods is that ours allows the registration network to aggregate the spatial information temporally, both forward and backward (see Figure 2(a)). The conventional approaches [9, 10, 15, 16, 24, 31] only conduct between two adjacent frames, which ignore the relationship of long-term dependency of the cardiac motion (see Figure 2(b)). Secondly, despite the large deformation through the path of diffeomorphism, the space of periodically specific human cardiac motion variation is bounded. Though methods [14, 15, 16] use Lagrangian strain to formulate the continuous dynamic of cardiac motion, however, without considering the motion consistency between two adjacent state spaces, the Lagrangian strain is prone to error accumulation and degrade the tracking performance. To address this problem, we take the simple yet efficient Bayesian approach, which employs the Gaussian Process (GP) to model cardiac motion dynamics in the compact feature space, predicting more consistent motion fields over dynamics parameters. Our proposed GPTrack can also be easily extended to other modalities or motion-tracking tasks, such as 3D Echocardiogram videos and 4D cardiac MRI.

In this paper, we follow the research [41] that employs the recursive manner in transformer for sequential data. As shown in Figure 3 left, the GPTrack pipeline comprised the GPTrack layer for feature extraction, the Gaussian Process (GP) layer for modelling the cardiac motion dynamics, and the Decoder for motion field estimation. Given the sequential 4D inputs $\{x_t\}_{t=1}^{T}, x_t \in \mathbb{R}^{H \times W \times D \times 1}$, where $H, W, D, T$ denote the height, width, depth, length of the input. For each $x_t$, we first decompose it to $P$ non-overlapping patches of shape $p \times p \times p$, where $P = \frac{H}{p} \times \frac{W}{p} \times \frac{D}{p}$ and $p, \frac{H}{p}, \frac{W}{p}, \frac{D}{p} \in \mathbb{Z}^{+}$. We then embed each patch as a feature with $C$ channels via embedding layers and disentangle patches with the dimension of $\mathbb{R}^{P \times C}$ from $x_t$. The GPTrack layer then takes the $x$, both forward and backward hidden states $\vec{h}, \overleftarrow{h} \in \mathbb{R}^{P \times C}$ as the input, then predicts the motion field $\phi$ via the decoder after the GP layer. Note that the initial hidden states of forward $\vec{h}_0$ and backward $\overleftarrow{h}_0$ are set as zero.

## 3.3 Bidirectional Forward-Backward Recursive Cell

The GPTrack layer consists of two independent GPTrack cells that respond to forward and backward computation. Similar to the [41, 43], adapting the hidden state to maintain and aggregate the sequential information allows the input with variable length. Meanwhile, the conventional convolutional neural network or vision in the transformer-based method is limited by the fixed input length. Furthermore, medical images such as Echocardiogram videos usually consist of hundreds of frames that cover multiple heartbeat cycles. Hence, parallel computing all frames requires a large amount of computational consumption, which hinders the application in real scenarios limited by low-computational devices. To this end, as shown in Tables 1 and 2, our GPTrack is able to formulate the variable temporal information while maintaining the comparable computational cost.

Using the forward GPTrack cell shown in the right of Figuer 3 as an example, the $x_t$ and $\vec{h}_{t-1}$ followed by the addition of learnable position encoding $\text{pos}_t \in \mathbb{R}^{P \times C}$ are respectively normalized by Layer Normalization. The linear self-attention then computes the attentive weight $A_t \in \mathbb{R}^{P \times C}$ of combined $x_t$ and $\vec{h}_{t-1}$. The above operations can be formulated as follows:

$$A_t = (\delta(\mathcal{W}_Q x) + 1)(\delta(\mathcal{W}_K x) + 1)^\mathsf{T} \mathcal{W}_V x; \ x = \text{LN}(x_t + \text{pos}_t) \oplus \text{LN}(\vec{h}_{t-1} + \text{pos}_t) \in \mathbb{R}^{P \times 2C}, \ (2)$$

where $\mathcal{W}_Q, \mathcal{W}_K, \mathcal{W}_V \in \mathbb{R}^{2C \times C}$ are learnable weights of different projection layers named query, key and value, $\delta(\cdot)$ represent the exponential linear units $elu(\cdot)$ [42], $\text{LN}(\cdot)$ and $\oplus$ are the layer normalization and the concatenation operation, respectively.

In order to raise the descendant hidden state $\vec{h}_t$ that aggregates information before $t + 1$ moment. The attentive weight $A_t$ then conducts the element-wise addition with ancestral positional encoded $\vec{h}_{t-1}$. Additionally, the $A_t$ takes the positional encoded $x_t$ as the residual connection and applies the addition operation. The Feed Forward Network denoted as $\text{FFN}(\cdot)$, is then introduced to output the feature of $t$-th moment. The formulation can be written as follows:

$$\vec{h}_t = A_t + \text{LN}(\vec{h}_{t-1} + \text{pos}_s) \in \mathbb{R}^{P \times C}, \ \vec{f}_t = \text{FFN}(\text{LN}(A_t + \text{LN}(x_t + \text{pos}_s))) \in \mathbb{R}^{P \times C}. \quad (3)$$

In the Bidirectional Forward-Backward Recursive Cell, both forward and backward share the same computation processes through the GPTrack cell. The only difference between the two directions is that the forward process starts from the first moment $x_0$ of input while the backward starts from the last moment $x_T$. Hence, the feature $f_t$ in $t$-th moment is formulated as $f_t = \vec{f}_t + \overleftarrow{f}_t$, $f_t \in \mathbb{R}^{P \times C}$, where $\vec{f}_t$ aggregate the forward information from 0 to $t$, while the $\overleftarrow{f}_t$ aggregate the backward information from the last frame to the $t$-th frame.

## 3.4 Gaussian Process in Cardiac Motion Tracking

The primary objective of integrating the Gaussian Process (GP) is to establish a probabilistic prior in the latent space that incorporates prior knowledge. Specifically, it posits that cardiac motion across different individuals within the same region should yield similar motion fields in latent space encodings. Furthermore, as illustrated in Figure 1, cardiac structures within an individual that are spatially distant or exhibit motions in opposite directions should consistently adhere to the periodic pattern of motion.

Initially, we define a covariance (kernel) function for the GP layer as depicted in Figure 3. We design the prior for the latent space processes to be stationary, mean square continuous, and differentiable in sequential motion fields. This design stems from our expectation that the latent functions should model cardiac motion more prominently than visual features. Consequently, we anticipate the latent space to manifest continuous and relatively smooth behaviour. To this end, we employ the isotropic and stationary Matern kernel (refer to Equation 4) to fulfil the required covariance function structure:

$$\kappa(x_t, x_{t-1}) = \sigma \frac{2^{1-\nu}}{\Gamma(\nu)} (\sqrt{2\nu} \frac{D(x_t, x_{t-1})}{l})^\nu K_\nu(\sqrt{2\nu} \frac{D(x_t, x_{t-1})}{l}), \qquad (4)$$

where $\nu, \sigma, l > 0$ are the smoothness, magnitude and length scale parameters, $K_\nu$ is the modified Bessel function, and $D(\cdot, \cdot)$ denotes the distance metric between features of two consecutive motion fields. Our goal is to formulate cardiac motion as robust prior knowledge applicable to unseen data. To address this, we propose a position-related distance measurement.

As outlined in section 3.3 and referenced in [44], we utilize a learnable parameter $\dot{\text{pos}} \in \mathbb{R}^{P \times C}$ as the spatial positional encoding for each region, which provides relative or absolute positional information about the decomposed patches. To capture the periodic temporal positional information of cardiac motion, distinct from the spatial encoding $\dot{\text{pos}}$, we apply sine and cosine functions of various frequencies as temporal encoding $\{\tilde{\text{pos}}_t\}_{t=1}^T, \tilde{\text{pos}}_t \in \mathbb{R}^{P \times C}$ for each moment. The overall positional encoding at moment $t$ is formulated as:

$$\text{pos}_t = \dot{\text{pos}} \cdot \tilde{\text{pos}}_t \in \mathbb{R}^{P \times C}, \ \text{where } \tilde{\text{pos}}_t(t, i) = \mathbb{I}_{(i=2k)} \sin(t \cdot n^{-\frac{2k}{C}}) + \mathbb{I}_{(i=2k+1)} \cos(t \cdot n^{-\frac{2k}{C}}). \ (5)$$

The $i$ denotes the $i$-th position of $C$ channels, and $n$ is the scaling factor. We assign independent GP priors to all values in $\{z_t\}_{t=1}^T, z_t \in \mathbb{R}^{P \times C}$ to disseminate temporal information between frames

in the sequence. During this process, we regard the sequential output $\{f_t\}_{t=1}^T$ of GPTrack as noise-corrupted versions of the ideal latent space encodings, formulating the inference as the following GP regression model with noise observations:

$$z_t \sim \text{GP}\left(\mu(\text{pos}_t), \kappa\left(\text{pos}_{t-1}, \text{pos}_t\right)\right), \; f_t = z_t + \epsilon_t, \; \epsilon_t \sim \mathcal{N}(0, \sigma^2),$$

where $\sigma^2$ is the noise variance of the likelihood model set as the learnable parameter in GPTrack. The above Gaussian process can be treated as the temporal sequence with intrinsic Markov property, and we adopt the methodology of connecting the Gaussian process with state space model [45] to decrease its computational complexity from $O(T^3)$ to $O(T)$, where $T$ is the number of time step. Concretely, the Gaussian process of Equation 6 corresponds to the following linear stochastic differential equation:

$$\frac{d}{dt}\mathbf{z}(t) = \mathbf{A}\mathbf{z}(t) + \mathbf{b}w(t), \; f(t) = \mathbf{h}^{\mathsf{T}}\mathbf{z}(t) + \epsilon(t), \; \epsilon(t) \sim \mathcal{N}(0, \sigma^2), \tag{6}$$

with the solution as:

$$\mathbf{z}(t) = \exp^{(t-r)\mathbf{A}}\mathbf{z}(r) + \int_r^t \exp^{(t-s)\mathbf{A}} \mathbf{b}w(s)ds, \; \forall r < t,$$
$$f(t) = \mathbf{h}^{\mathsf{T}}\mathbf{z}(t) + \epsilon(t), \; \epsilon(t) \sim \mathcal{N}(0, \sigma^2), \tag{7}$$

where $\mathbf{z}(t) := (z(t), \frac{d}{dt}z(t))$, $w(t)$ is the zero-mean Gaussian random process, and $\mathbf{h} := (0, 1)^{\mathsf{T}}$ is used for modelling observation model. The state transition matrix (vector) $\mathbf{A}$ and $\mathbf{b}$ can be calculated from the covariance function $\kappa$ of the Gaussian process. For the Matern kernel shown in Equation 4, we take $\nu = \frac{3}{2}$, and corresponding state transition matrix $\mathbf{A}$ and vector $\mathbf{b}$ [46] read as:

$$\mathbf{A} = \begin{pmatrix} 0, & 1 \\ -\frac{3}{l^2}, & -\frac{2\sqrt{3}}{l} \end{pmatrix}, \; \mathbf{b} = (0, 1)^{\mathsf{T}}. \tag{8}$$

Then we can discretize Equation 7 and get its weakly equivalent state-space model of Equation as:

$$\mathbf{z}_t = \mathbf{\Phi}_t \, \mathbf{z}_{t-1} + \mathbf{n}_t, \; f_t = \mathbf{h}^{\mathsf{T}}\mathbf{z}_t + \epsilon_t, \; \epsilon(t) \sim \mathcal{N}(0, \sigma^2), \; t = 1, \dots, T, \tag{9}$$

where $\mathbf{\Phi}_t = \exp^{D(\text{pos}_t, \text{pos}_{t-1})\mathbf{A}}$, $\mathbf{n}_t \sim \mathcal{N}(0, \mathbf{\Phi}_t \mathbf{b}Q_w(t-1, t)\mathbf{b}^{\mathsf{T}}\mathbf{\Phi}_t^{\mathsf{T}})$, and $Q_w(t, t-1)$ is the co-variance of $w(t)$. Given the initial value $\mathbf{z}_0 \sim \mathcal{N}(\boldsymbol{\mu}_0, \boldsymbol{\Sigma}_0)$ with $\boldsymbol{\mu}_0 = 0$ and $\boldsymbol{\Sigma}_0 = \text{diag}(\frac{\sigma^2}{2}, \frac{3\sigma^2}{l^2})$, we can sequentially calculate the posterior distribution $\mathbf{z}_t|f_{1:t-1} \sim \mathcal{N}(\overline{\boldsymbol{\mu}}_t, \overline{\boldsymbol{\Sigma}}_t)$ and $\mathbf{z}_t|f_{1:t} \sim \mathcal{N}(\boldsymbol{\mu}_t, \boldsymbol{\Sigma}_t)$ using update criterion of Kalman filter for state space model [47] as:

$$\overline{\boldsymbol{\mu}}_t \leftarrow \mathbf{\Phi}_t\overline{\boldsymbol{\mu}}_{t-1}, \qquad\qquad \overline{\boldsymbol{\Sigma}}_t \leftarrow \mathbf{\Phi}_t\overline{\boldsymbol{\Sigma}}_{t-1}\mathbf{\Phi}_t^{\mathsf{T}} + \boldsymbol{\Sigma}_0 - \mathbf{\Phi}_t\boldsymbol{\Sigma}_0\mathbf{\Phi}_t^{\mathsf{T}},$$
$$\boldsymbol{\mu}_t \leftarrow \overline{\boldsymbol{\mu}}_t + \mathbf{k}_t(f_t - \mathbf{h}^{\mathsf{T}}\overline{\boldsymbol{\mu}}_t), \;\; \boldsymbol{\Sigma}_t \leftarrow \overline{\boldsymbol{\Sigma}}_t - \mathbf{k}_t\mathbf{h}^{\mathsf{T}}\overline{\boldsymbol{\mu}}_t, \; t = 1, \dots, T, \tag{10}$$

where $\mathbf{k}_t := \frac{\overline{\boldsymbol{\Sigma}}_t\mathbf{h}}{\mathbf{h}\overline{\boldsymbol{\Sigma}}_t\mathbf{h}+\sigma^2}$ is the optimal Kalman gain at time $t$. The output of the GP layer in $t$-th moment thus can be formulated as $z_t^{GP} = \text{ReLU}(\mathbf{k}_t z_t)$, where ReLU is the activation function. In the final, the $t$-th motion field $\theta_t$ is obtained by decoder from the $z_t^{GP}$.

### 3.5 Overall Loss of Tracking a Time Sequence of Cardiac Motion

The decoder takes the Gaussian-process coding $\{z_t^{GP}\}_{t=1}^T$ as velocity filed to composite the diffeomorphic motion field $\phi$ according to the criterion of Section 3.1. Here, we adopt the training loss of [15, 16], which minimizes the integration of four components summarized as follows: **a)** Dissimilarities of tracking results between adjacent states from both forward and backward; **b)** Smoothness of motion fields between adjacent states from both forward and backward; **c)** Dissimilarities of tracking results between the start state and each state; **d)** Smoothness of motion fields between the start state and each state. The overall loss function $\mathcal{L}$ is formulated as:

$$\sum_{t=1}^{T-1}[\underbrace{\mathcal{L}_{kl}(x_t, x_{t+1})}_{\textbf{a)}} + \alpha_1(\underbrace{\mathcal{L}_{sm}(\phi_{t:t+1}) + \mathcal{L}_{sm}(\phi_{t+1:t})}_{\textbf{b)}}) + \alpha_2 \underbrace{\mathcal{L}_{nc}(x_{t+1}, x_1 \circ \phi_{0:t+1})}_{\textbf{c)}} + \alpha_3 \underbrace{\mathcal{L}_{sm}(\phi_{1:t+1})}_{\textbf{d)}}],$$

where $\alpha_1$, $\alpha_2$ and $\alpha_3$ are loss weights, and $\phi_{t_1:t_2}$ is the motion field from state $t_1$ to $t_2$. $\mathcal{L}_{kl}(x_t, x_{t+1}) = \mathbb{KL}(q(z_t^{GP}|x_t; x_{t+1})||p(z_t^{GP}|x_t; x_{t+1})) + \mathbb{KL}(q(z_t^{GP}|x_{t+1}; x_t)||p(z_t^{GP}|x_{t+1}; x_t))$ is the summation of forward and backward VAE losses with latent coding $z_t^{GT}$, posterior distribution $q$ and conditional distribution $p$, $L_{nc}$ is the negative normalized local cross-correlation metric, and $\mathcal{L}_{sm}(\phi) = ||\nabla\phi||_2^2$ is the $\ell_2$-total variation metric.

# 4 Experiment

## 4.1 Datasets

**CardiacUDA [17].** The CardiacUDA dataset collected from two medical centers consists of 314 echocardiogram videos from patients. The video is scanned from the apical four-chamber heart (A4C) view. In this paper, we conduct training and validation in the A4C view that consists of 314 videos with 5 frame annotations in the Left/Right Ventricle and Atrium (LV, LA, RV, RA). For testing, we report our results in 10 videos with full annotation provided by the CardiacUDA.

**CAMUS [18].** The CAMUS dataset provides pixel-level annotations for the left ventricle, myocardium, and left atrium in the Apical two-chamber view, which consists of 500 echocardiogram videos in total. There are 450 subjects in the training set with 2 frames annotated in the Left Ventricle (LV), Left Atrium (LA) and Myocardium (Myo) in the end-diastole (ED) and end-systole (ES) of the heartbeat cycle. The remaining 50 subjects without any annotation masks belong to the testing set.

**ACDC [19].** The ACDC dataset consists of 100 4D temporal cardiac MRI cases. All data provide the segmentation annotations corresponding with the Left Ventricle (LV), Left Atrium (LA) and Myocardium (Myo) in the end-diastole (ED) and end-systole (ES) during the heartbeat cycle.

## 4.2 Implementation Details

**Training.** We trained the model using the Adam optimizer with betas equal to 0.9 and 0.99. The training batch size of the model was set to 1. We trained for a total of 1000 epochs with an initial learning rate of $5e^{-4}$ and decay by a factor of 0.5 in every 50 epochs. During training, for CardiacUDA [17] and CAMUS [18], we resized each frame to $384 \times 384$ and then randomly cropped them to $256 \times 256$. All frames were normalized to [0,1] during training. In temporal augmentation of datasets [17, 18], we randomly selected 32 frames from an echocardiogram video with a sampling ratio of either 1 or 2. For ACDC [19], we resampled all scans with a voxel spacing of 1.5 × 1.5 × 3.15mm and cropped them to $128 \times 128 \times 32$, normalized the intensity of all images to [-1, 1]. For spatial data augmentation of all datasets, we randomly applied flipping, rotation and Gaussian blurring. In CardiacUDA, we split the dataset into $8:2$ for training and validation. During testing, we reported results in 10 fully annotated videos. In the CAMUS [18] dataset, videos without annotation are used for only training, while we randomly split the remaining 450 annotated videos into 300/50/100 for training, validation and testing. In the ACDC [19], following the [11, 12], we split the training set in the ratio of 90 and 10 for training and testing. The reproduced methods strictly follow the official code and the description in the paper. For all experiments, We use Intel(R) Xeon(R) Platinum 8375C with $1\times$ RTX3090 for both training and inference. All reproduced methods strictly followed the training settings with their original paper in the same experimental environment.

**Inference.** For CardiacUDA and CAMUS, we resized videos to $384 \times 384$, cropped to $256 \times 256$ in central and normalized to [0,1]. We sample 32 frames that cover the segmentation annotation. When the sequence has more than 32 frames, the extra frames will be removed from the sequence, except for the first and the last one. The ACDC dataset remains the same sampling strategy as training in the inference stage, without any argumentation except for normalizing intensity to [-1,1].

**Evaluation Metrics.** For the evaluation of the quality of registered target frames, we follow [12] to use the Peak Signal-to-Noise Ratio (PSNR) and Structural Similarity Index (SSIM) [48] to measure whether the Lagrangian motion field is accurately estimated between the first frame and the following wrapped frames. We also use the Dice [49] score to measure the discrepancy between tracked and ground-truth cardiac segmentation. For CardiacUDA [17], only the first frame and corresponding segmentation are provided for tracking the following 32 frames, then report the averaged results by the above metrics in these 32 frames. For CAMUS [18] and ACDC [19], frame and segmentation of the ED stage are used to track the go-after frames, and we report all the metrics in the ES stage. We evaluate diffeomorphic property by computing the percentage of non-positive values of the Jacobian determinant $det(J_\phi) \leq 0$ (%) on the Lagrangian motion field. In order to access the evaluation of comparing the physiological plausibility following the [50, 51], we also compute the mean absolute difference between the 1 and Jacobian determinant ($||J_\phi| - 1|$) over the tracking areas. For a fair comparison, we evaluate the computational efficiency and report the computational time in seconds (Times), the parameter quantities in millions (Params), and the tera-floating point operations per second (TFlops). *We also provide the result evaluated by Hussdorf Distance (HD) in Tables B1, B2 and B3 of Appendix Section B.*

Table 1: The performance[1] of different registration methods in Cardiac-UDA dataset [17]. Results were reported in structures (RV, RA, LV, LA) and the overall averaged Dice score (Avg. %).

| 2D Methods (256×256) | LV ↑ | RV ↑ | LA ↑ | RA ↑ | Avg. ↑ | $\|J\|-1\|$ ↓ | $det(J_\phi)\leq 0$ ↓ | PSNR ↑ | SSIM ↑ | Times (s) ↓ | Params (M) ↓ | TFlops ↓ |
|---|---|---|---|---|---|---|---|---|---|---|---|---|
| | | | | | Non-rigid Registration | | | | | | | |
| LDDMM [6] | $69.44_{\pm6.9}$ | $70.61_{\pm5.3}$ | $57.03_{\pm12}$ | $70.78_{\pm5.3}$ | $69.22_{\pm5.2}$ | $13.12_{\pm11.05}$ | $25.67_{\pm23.41}$ | $26.45_{\pm2.7}$ | $76.44_{\pm2.4}$ | $^*177.9_{\pm2.3}$ | - | - |
| RDMM [8] | $70.50_{\pm7.3}$ | $71.12_{\pm6.3}$ | $57.10_{\pm12}$ | $72.22_{\pm6.0}$ | $70.84_{\pm6.3}$ | $5.102_{\pm1.067}$ | $8.602_{\pm6.350}$ | $26.80_{\pm2.6}$ | $76.92_{\pm1.9}$ | $^*241.0_{\pm3.5}$ | - | - |
| ANTs (SyN) [24] | $73.51_{\pm6.6}$ | $74.12_{\pm5.7}$ | $60.49_{\pm14}$ | $74.69_{\pm4.6}$ | $73.71_{\pm5.8}$ | $16.09_{\pm8.031}$ | $40.06_{\pm28.56}$ | $27.96_{\pm2.4}$ | $76.52_{\pm2.5}$ | $^*156.4_{\pm4.1}$ | - | - |
| | | | | | Deep Learning Based Registration | | | | | | | |
| VM-SSD [10] | $74.26_{\pm8.3}$ | $74.85_{\pm5.2}$ | $66.78_{\pm18}$ | $76.24_{\pm7.4}$ | $75.86_{\pm4.2}$ | $0.374_{\pm0.021}$ | $0.262_{\pm0.305}$ | $29.01_{\pm2.5}$ | $75.89_{\pm1.8}$ | $0.011_{\pm0.0}$ | 0.118 | 0.010 |
| VM-NCC [10] | $74.04_{\pm7.2}$ | $76.20_{\pm5.9}$ | $67.54_{\pm14}$ | $77.36_{\pm4.2}$ | $76.51_{\pm4.2}$ | $0.685_{\pm0.052}$ | $0.905_{\pm1.229}$ | $28.53_{\pm2.5}$ | $75.77_{\pm2.3}$ | $0.011_{\pm0.0}$ | 0.118 | 0.010 |
| SYMNet [36] | $75.21_{\pm7.5}$ | $75.33_{\pm6.1}$ | $69.67_{\pm11}$ | $77.78_{\pm5.5}$ | $76.60_{\pm4.2}$ | $0.454_{\pm0.048}$ | $0.631_{\pm0.108}$ | $28.56_{\pm2.5}$ | $76.87_{\pm2.0}$ | $0.101_{\pm0.0}$ | 0.449 | 0.125 |
| VM-DIF [9] | $73.53_{\pm7.5}$ | $76.37_{\pm5.6}$ | $68.10_{\pm15}$ | $78.55_{\pm6.1}$ | $76.83_{\pm5.0}$ | $0.387_{\pm0.066}$ | $0.437_{\pm0.508}$ | $28.80_{\pm2.2}$ | $76.87_{\pm1.8}$ | $\underline{0.011}_{\pm0.0}$ | $\underline{0.109}$ | $\underline{0.010}$ |
| Ahn SS, et al. [31] | $75.66_{\pm7.6}$ | $77.24_{\pm6.3}$ | $71.41_{\pm17}$ | $79.20_{\pm6.9}$ | $77.04_{\pm4.3}$ | $3.107_{\pm1.156}$ | $2.664_{\pm0.827}$ | $29.86_{\pm2.5}$ | $77.59_{\pm2.4}$ | $0.017_{\pm0.0}$ | 7.783 | 0.851 |
| DiffuseMorph [12] | $77.02_{\pm6.0}$ | $80.45_{\pm5.5}$ | $72.50_{\pm12}$ | $80.81_{\pm5.3}$ | $79.27_{\pm5.2}$ | $0.319_{\pm0.043}$ | $0.339_{\pm0.478}$ | $29.48_{\pm2.0}$ | $77.02_{\pm2.5}$ | $0.103_{\pm0.0}$ | 90.67 | 0.227 |
| DeepTag [15, 16] | $76.83_{\pm7.5}$ | $80.13_{\pm4.8}$ | $72.87_{\pm14}$ | $80.98_{\pm4.2}$ | $79.41_{\pm3.5}$ | $0.273_{\pm0.056}$ | $0.027_{\pm0.022}$ | $28.53_{\pm2.5}$ | $76.40_{\pm2.3}$ | $\mathbf{0.011}_{\pm0.0}$ | **0.107** | **0.010** |
| GPTrack-M (Ours) | $76.94_{\pm7.6}$ | $81.72_{\pm6.4}$ | $73.13_{\pm16}$ | $80.85_{\pm6.4}$ | $81.64_{\pm2.8}$ | $0.286_{\pm0.049}$ | $0.119_{\pm0.084}$ | $31.28_{\pm2.0}$ | $78.22_{\pm2.4}$ | $0.013_{\pm0.0}$ | 0.467 | 0.015 |
| GPTrack-L (Ours) | $77.07_{\pm8.0}$ | $82.57_{\pm7.1}$ | $73.11_{\pm15}$ | $\mathbf{81.24}_{\pm6.4}$ | $82.11_{\pm2.7}$ | $\mathbf{0.250}_{\pm0.044}$ | $\underline{0.019}_{\pm0.017}$ | $31.57_{\pm2.0}$ | $78.70_{\pm2.1}$ | $0.016_{\pm0.0}$ | 5.161 | 0.041 |
| GPTrack-XL (Ours) | $\mathbf{78.51}_{\pm7.9}$ | $\mathbf{82.48}_{\pm6.0}$ | $\mathbf{73.43}_{\pm12}$ | $\underline{81.20}_{\pm5.9}$ | $\mathbf{82.37}_{\pm2.7}$ | $0.279_{\pm0.085}$ | $0.027_{\pm0.023}$ | $\mathbf{32.03}_{\pm2.4}$ | $\mathbf{80.04}_{\pm2.4}$ | $0.026_{\pm0.0}$ | 7.536 | 0.053 |

Table 2: The performance[1] of different registration methods in ACDC [19] dataset. Results reported in structures (RV, LV, Myo) and overall averaged Dice score (Avg. %).

| 3D Methods (128×128×32) | RV ↑ | LV ↑ | Myo ↑ | Avg. ↑ | $\|J\|-1\|$ ↓ | $det(J_\phi)\leq 0$ ↓ | PSNR ↑ | SSIM ↑ | Times (s) ↓ | Params (M) ↓ | TFlops ↓ |
|---|---|---|---|---|---|---|---|---|---|---|---|
| | | | | Non-rigid Registration | | | | | | | |
| LDDMM [6] | $73.61_{\pm8.5}$ | $65.62_{\pm8.5}$ | $56.44_{\pm13}$ | $72.39_{\pm18}$ | $451.8_{\pm162.3}$ | $653.5_{\pm371.2}$ | $31.20_{\pm3.8}$ | $84.59_{\pm6.0}$ | $^*1533_{\pm8.4}$ | - | - |
| RDMM [8] | $76.43_{\pm7.8}$ | $69.50_{\pm9.1}$ | $62.19_{\pm14}$ | $75.51_{\pm12}$ | $144.2_{\pm63.67}$ | $266.0_{\pm165.3}$ | $31.66_{\pm3.9}$ | $84.36_{\pm5.4}$ | $^*1715_{\pm26}$ | - | - |
| ANTs (SyN) [24] | $75.30_{\pm7.4}$ | $66.92_{\pm8.6}$ | $58.03_{\pm11}$ | $74.64_{\pm13}$ | $15.82_{\pm22.30}$ | $57.26_{\pm37.74}$ | $30.92_{\pm3.6}$ | $84.26_{\pm5.6}$ | $^*1166_{\pm16}$ | - | - |
| | | | | Deep Learning Based Registration | | | | | | | |
| VM-SSD [10]] | $79.83_{\pm7.1}$ | $74.27_{\pm9.0}$ | $64.44_{\pm15}$ | $77.56_{\pm12}$ | $3.144_{\pm2.242}$ | $4.602_{\pm3.485}$ | $32.61_{\pm3.7}$ | $83.88_{\pm5.2}$ | $0.015_{\pm0.0}$ | 0.327 | 0.767 |
| VM-NCC [10] | $81.60_{\pm6.5}$ | $77.00_{\pm8.6}$ | $67.90_{\pm13}$ | $79.90_{\pm11}$ | $0.260_{\pm0.070}$ | $0.079_{\pm0.058}$ | $34.68_{\pm3.3}$ | $85.01_{\pm5.5}$ | $0.015_{\pm0.0}$ | 0.327 | 0.767 |
| VM-DIF [9] | $81.50_{\pm6.6}$ | $75.50_{\pm9.2}$ | $65.90_{\pm14}$ | $78.90_{\pm12}$ | $0.286_{\pm0.074}$ | $0.083_{\pm0.063}$ | $33.48_{\pm3.5}$ | $84.22_{\pm5.1}$ | $\underline{0.015}_{\pm0.0}$ | **0.327** | 0.767 |
| SYMNet [36] | $80.46_{\pm6.4}$ | $77.81_{\pm9.4}$ | $66.22_{\pm14}$ | $79.47_{\pm13}$ | $0.341_{\pm0.062}$ | $0.121_{\pm0.054}$ | $32.91_{\pm3.5}$ | $83.55_{\pm4.9}$ | $0.414_{\pm0.0}$ | 1.124 | 0.226 |
| NICE-Trans [52] | $79.97_{\pm6.0}$ | $78.55_{\pm8.1}$ | $67.02_{\pm11}$ | $79.66_{\pm10}$ | $0.278_{\pm0.071}$ | $0.093_{\pm0.044}$ | $33.08_{\pm3.0}$ | $83.88_{\pm4.7}$ | $0.486_{\pm0.0}$ | 5.619 | 0.280 |
| DiffuseMorph [12] | $82.10_{\pm6.7}$ | $78.30_{\pm8.6}$ | $67.80_{\pm15}$ | $80.50_{\pm11}$ | $0.237_{\pm0.068}$ | $0.061_{\pm0.038}$ | $34.73_{\pm3.6}$ | $84.30_{\pm5.2}$ | $0.458_{\pm0.0}$ | $\underline{0.327}$ | 0.642 |
| CorrMLP [53] | $80.33_{\pm6.5}$ | $80.07_{\pm7.8}$ | $70.51_{\pm14}$ | $80.44_{\pm8.6}$ | $0.248_{\pm0.055}$ | $0.059_{\pm0.022}$ | $34.90_{\pm2.9}$ | $84.27_{\pm4.5}$ | $0.070_{\pm0.0}$ | 13.36 | 0.303 |
| DeepTag [15, 16] | $81.89_{\pm7.0}$ | $79.10_{\pm7.5}$ | $70.37_{\pm13}$ | $80.83_{\pm12}$ | $0.185_{\pm0.067}$ | $0.044_{\pm0.025}$ | $33.64_{\pm3.4}$ | $83.09_{\pm4.9}$ | $\underline{0.015}_{\pm0.0}$ | 0.362 | **0.113** |
| Transmatch [54] | $81.22_{\pm7.0}$ | $80.34_{\pm6.8}$ | $71.21_{\pm12}$ | $81.35_{\pm9.8}$ | $0.226_{\pm0.050}$ | $0.077_{\pm0.054}$ | $33.89_{\pm3.3}$ | $84.78_{\pm4.9}$ | $0.325_{\pm0.0}$ | 70.71 | 0.603 |
| FSDiffReg [11] | $82.70_{\pm6.1}$ | $80.90_{\pm7.7}$ | $72.40_{\pm12}$ | $82.30_{\pm9.6}$ | $0.214_{\pm0.026}$ | $0.054_{\pm0.026}$ | $35.34_{\pm3.5}$ | $85.85_{\pm5.2}$ | $1.106_{\pm0.0}$ | 1.320 | 0.855 |
| GPTrack-M (Ours) | $81.65_{\pm7.0}$ | $80.77_{\pm7.5}$ | $71.53_{\pm16}$ | $81.45_{\pm10}$ | $0.209_{\pm0.081}$ | $0.047_{\pm0.035}$ | $34.82_{\pm3.2}$ | $85.78_{\pm5.3}$ | $0.022_{\pm0.0}$ | 0.418 | $\underline{0.201}$ |
| GPTrack-L (Ours) | $82.78_{\pm5.6}$ | $81.16_{\pm6.8}$ | $71.71_{\pm14}$ | $82.38_{\pm11}$ | $0.182_{\pm0.072}$ | $0.035_{\pm0.022}$ | $34.99_{\pm3.0}$ | $85.62_{\pm4.9}$ | $0.023_{\pm0.0}$ | 0.942 | 0.204 |
| GPTrack-XL (Ours) | $\mathbf{82.91}_{\pm5.8}$ | $81.23_{\pm8.2}$ | $72.86_{\pm9.0}$ | $\mathbf{82.65}_{\pm10}$ | $\mathbf{0.178}_{\pm0.024}$ | $0.032_{\pm0.021}$ | $35.52_{\pm3.1}$ | $86.19_{\pm5.0}$ | $0.034_{\pm0.0}$ | 1.094 | 0.205 |

## 4.3 Results

**Result of 3D Echocardiogram Video.** In Table 1, we compare our method with state-of-the-arts 2D registration [9, 10, 12, 15, 31, 52] and non deep learning [6, 8, 24] methods in CardiacUDA dataset. In comparison to DeepTag [15], our GPTrack-XL reach $82.37\%$ and $12.75$ in DICE and HD scores with the best non-positive Jacobian determinant value, which denotes the learned motion field is smooth. The registration quality of our method also achieved the best with $32.03$ ($3.50 \uparrow$) and $80.04$ ($3.64 \uparrow$) in PSNR and SSIM compared to the second-best method, respectively. Though GP-

Table 3: The segmentation performance[1] of different cardiac structures in the CAMUS [18].

| Methods (256×256) | CAMUS (Dice%) | | | |
|---|---|---|---|---|
| | LV | LA | Myo | Avg. |
| VM-NCC [10] | $79.50_{\pm6.8}$ | $80.10_{\pm9.7}$ | $67.70_{\pm9.3}$ | $75.80_{\pm7.0}$ |
| SYMNet [36] | $85.24_{\pm7.0}$ | $82.77_{\pm9.6}$ | $76.36_{\pm6.2}$ | $81.84_{\pm4.5}$ |
| VM-SSD [10] | $86.30_{\pm6.7}$ | $85.20_{\pm9.3}$ | $77.90_{\pm6.9}$ | $83.10_{\pm4.5}$ |
| Ahn SS, et al. [31] | $86.42_{\pm7.2}$ | $83.95_{\pm8.9}$ | $75.68_{\pm8.4}$ | $82.33_{\pm4.1}$ |
| DiffuseMorph [11] | $85.76_{\pm5.9}$ | $84.49_{\pm8.7}$ | $76.65_{\pm6.3}$ | $83.57_{\pm4.5}$ |
| VM-DIF [9] | $\underline{87.70}_{\pm6.0}$ | $85.40_{\pm10}$ | $\mathbf{80.40}_{\pm6.3}$ | $84.50_{\pm3.7}$ |
| DeepTag [15] | $87.60_{\pm4.5}$ | $\underline{87.90}_{\pm6.1}$ | $79.00_{\pm6.8}$ | $\underline{84.80}_{\pm5.1}$ |
| GPTrack-XL (Ours) | $\mathbf{88.63}_{\pm4.6}$ | $\mathbf{89.13}_{\pm8.0}$ | $\underline{80.37}_{\pm7.3}$ | $\mathbf{85.29}_{\pm4.2}$ |

Track introduces more learnable parameters and requires a small amount of additional computation, slightly increases inference time and TFlops compared to DeepTag [15] and VM-DIF [9]. GPTrack surpasses all other methods in the registration and verifies the necessity of formulating a strong temporal relationship among frames by using the recursive manner. Tables 1 and 3 show registration results of cardiac structures (LV, RV, LA, RA, Myo). Compared to other methods, our GPTrack can outperform existing baseline methods by a substantial margin. In areas such as the left atrium (LA) and left ventricular (LV), which usually cause larger deformation, our method can also provide better alignment than other approaches.

---

[1]Segmentation results are reported in the Dice score (%). **Bold**, underline denote the best results and the second best performance. The superscript * indicates computational time reported in only CPU implementation. We use the t-test for statistical significance analysis, where the p-value between the two methods is $p < 0.05$, indicating statistically significant improvements

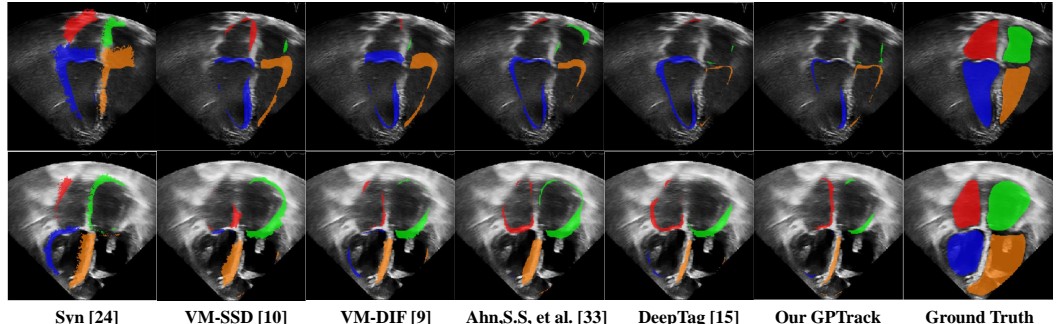

| Syn [24] | VM-SSD [10] | VM-DIF [9] | Ahn,S.S, et al. [33] | DeepTag [15] | Our GPTrack | Ground Truth |

Figure 4: The visualization in 3D Echocardiogram video of motion tracking error. We visualised the last frame of tracking result and ground truth from 32 consecutive frames in CardiacUDA [17]. Colours Red, Blue, Green and Orange denote cardiac structures RA, RV, LV and LA, respectively.

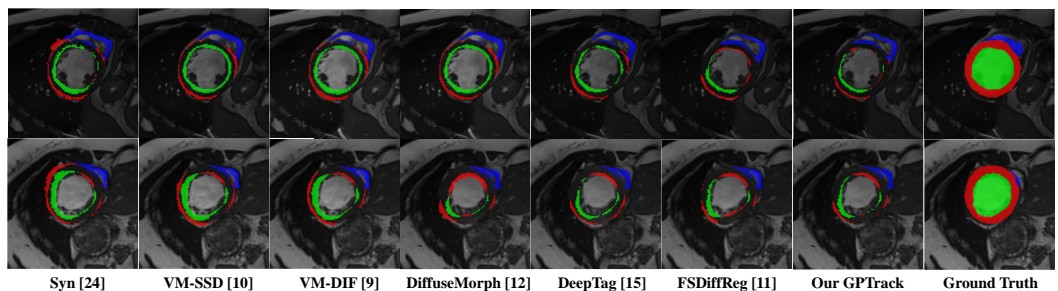

| Syn [24] | VM-SSD [10] | VM-DIF [9] | DiffuseMorph [12] | DeepTag [15] | FSDiffReg [11] | Our GPTrack | Ground Truth |

Figure 5: The visualization in 4D Cardiac MRI of motion tracking error. We visualised the result of the last frame tracking from ED to ES and corresponding ground truth in ACDC [17]. Colours Red, Blue, and Green denote cardiac structures MYO, LA, and LV, respectively.

As shown in Figure 4, the tracking error of our GPTrack and other methods show a significant difference in LA (Labelled by Orange Colour) and LV (Labelled by Green Colour) when compared to the ground truth. The methods [24, 10, 9, 31] present inaccuracy tracking results due to lack of the constraints on the consecutive motion and ignore the long-term temporal information. These results further verify that our specifically designed GPTrack for modelling motion patterns is more suitable for cardiac motion tracking on echocardiography.

**Result of 4D Temporal Cardiac MRI Dataset.** We compare our GPTrack method against with the state-of-the-art deep learning based methods [9, 10, 11, 12, 15] and different Non-rigid approaches [6, 8, 24]. As illustrated in Table 2, our GPTrack-XL achieves the best average DICE score of 82.65 compared to FSDiffReg [11] with 82.30. In registration quality, our GPTrack-XL reaches the highest scores, 31.52 and 86.19, in PSNR and SSIM, respectively. Moreover, our Jacobian determinant on deformation fields shows numbers comparable to other methods with the diffeomorphic constraint. All results are based on our fast and lightweight model, reducing around 96.93% inference time, 17.2% model parameters and 76.02% computational consumption (TFlops) compared to the second-best performance. In comparison to the diffusion-based method [12, 11], which requires enormous computation that hinders real-time inference and is nearly impossible to deploy in real scenarios, the GPTrack preserve light-weight and considerable performance by formulating cardiac motion patterns as the Gaussian process latent coding and bidirectionally understand the cardiac motion. The visualization result in Figure 5 also indicates our GPTrack can achieve better tracking accuracy.

## 4.4 Ablation Study

**The Scale of Model Hyper-parameters.** Table 4 shows the settings of the 2D/3D GPTrack-M/L/XL. For the 3D echocardiogram video dataset and 4D cardiac MRI dataset, the GPTrack with different scales has different patch sizes and dimension numbers. Referring to the result performed by Tables 3, 1 and 2, the registration result can be boosted by increasing the layers number and dimension size of GPTracks according to different requirements.

Table 4: The ablation study of the different configurations of our 2D / 3D GPTracks (M, L, XL).

| Model | Layer | Patch Size | Dim | GFlops | Param (M) |
|---|---|---|---|---|---|
| GPTrack-M | 2 / 2 | 16 / 8 | 64 / 32 | 1.54 / 20.1 | 0.467 / 0.418 |
| GPTrack-L | 2 / 2 | 16 / 8 | 256 / 64 | 4.18 / 20.4 | 5.161 / 0.942 |
| GPTrack-XL | 4 / 4 | 16 / 8 | 256 / 64 | 5.39 / 20.5 | 7.536 / 1.094 |

Table 5: Ablations of Bi-Directional (Bi-direct.) and Gaussian Process (GP) of **GPTrack-XL**.

| Bi-direct. | GP | Dice.Avg | $det(J_\phi) \leq 0$ |
|---|---|---|---|
| ✗ | ✗ | $79.83_{\pm 3.1}$ | $0.048_{\pm 0.040}$ |
| ✗ | ✓ | $80.81_{\pm 2.8}$ | $0.052_{\pm 0.047}$ |
| ✓ | ✗ | $81.21_{\pm 2.7}$ | $0.039_{\pm 0.031}$ |
| ✓ | ✓ | $\mathbf{82.37}_{\pm 3.2}$ | $\mathbf{0.027}_{\pm 0.023}$ |

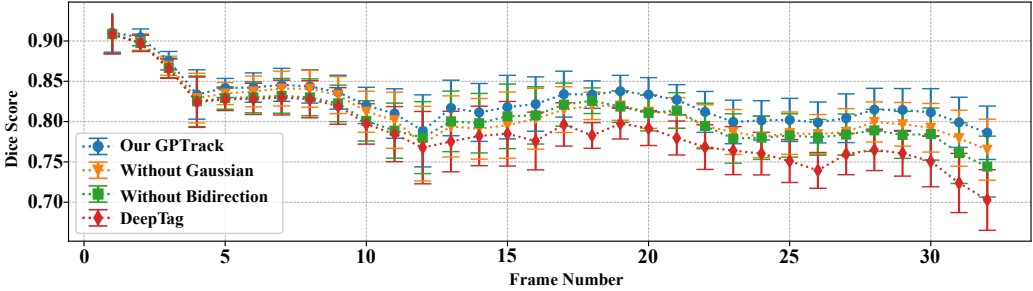

Figure 6: The Tracking Error performed by different methods in 32 consecutive frames of CardiacUDA [17] full annotated set.

**The Ablation of Bidirectional Recursive Manner and Gaussian Process.** Table 5 illustrates the Bi-directional layer in GPTracks can better formulate both forward and backward motion fields by aggregating the temporal information of the cardiac heartbeat cycle. The Gaussian Process models cardiac motion with strong prior knowledge from data, which makes more accurate predictions of the deformation field. The Figure 6 illustrate the tracking error from 1-st to 32-th frame in CardiacUDA [17] full annotated set. Our GPTrack and the DeepTag [15] both use the Lagrangian strain. However, the accuracy degrades significantly without aggregating the temporal information and GP when the input length increases. As shown in Figure 6, the tracking error after the 20-th frame becomes larger by using only Lagrangian strain, while introducing GP and bidirectional recursive methods can efficiently eliminate the tracking error by predicting more accurate motion fields.

## 5 Conclusion and Limitation

In this paper, we proposed a new framework named GPTrack to improve cardiac motion tracking accuracy. GPTrack innovatively aggregates both forward and backward temporal information by using the bidirectional recurrent transformer. Furthermore, we introduce the Gaussian Process to model the variability and predictability of cardiac motion. In experiments, our framework demonstrates the state-of-the-art in 3D echocardiogram and 4D cardiac MRI datasets. The limitation of our framework is that we use positional encoding as the prior knowledge of the cardiac motion, which may degrade the tracking performance in out-of-domain datasets. In our future work, we will build a more robust representation of cardiac motion and further our work across different medical domains. *We also provide the illustration of Broader Impacts, please see Section A2 in Appendix.*

## Acknowledgements

This work was partially supported by grants from the National Natural Science Foundation of China (Grant No. 62306254), the Hetao Shenzhen-Hong Kong Science and Technology Innovation Cooperation Zone (Project No. HZQB-KCZYB-2020083), and the Research Grants Council of the Hong Kong Special Administrative Region, China (Project Reference Number: T45-401/22-N).

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

# Appendix

## A1    Difference Between Optical Flow and Diffeomorphism Mapping

Optical flow (OF) based methods, often applied in object tracking of video sequences, have been explored by [3, 4, 5]. However, their effectiveness in medical imaging is limited due to challenges in accommodating large deformations and the inherent low quality of certain medical imaging modalities [4, 55], such as echocardiogram videos.

With the progression of deep learning, neural networks have also been employed to predict optical flow, which is crucial for predicting dynamic motion trajectories in video sequences. Notable implementations include FlowNet [37], iterative methods by [39], and self-supervised learning approaches by [38] and [56]. However, while supervised methods require a ground truth annotation for training cost functions [37, 39, 40, 57, 58], unsupervised approaches depend on photometric loss to ensure motion consistency [38, 56, 59], which can be challenging to obtain in medical images.

Last but not least, OF-based methods do not necessarily preserve topology, non-globally one-to-one (objective) smooth and continuous mapping with derivatives that are invertible. In cardiac motion tracking, we consider the deformation in each point of the adjacent frame to remain one-on-one mapping and be invertible for forward and backward deformation fields. Directly using the OF-based method to predict the motion field of cardiac motion may lead to incorrect estimation.

## A2    Broader Impacts

Our work focuses on cardiac motion tracking with an unsupervised framework named GPTrack. The GPTrack framework has the potential to support medical imaging physicians, such as radiologists and sonographers, in observing the cardiac motion of patients. This is a fundamental task for assessing cardiac function, and we are able to provide decision support and improve analysis efficiency and analysis reproducibility in clinical scenarios.

Moreover, our new framework illustrates that cardiac motion can be formulated as strong prior knowledge, which is able to be utilised to enhance tracking accuracy. Also, our work presents several advantages that help us make progress towards these benefits, which improve the performance of automated motion field estimation algorithms. The method not only improves the precision of motion tracking and segmentation in both 3D and 4D medical image modailites [17, 18, 19, 60, 61, 62] but also provides a comprehensive observation of motion information to radiologists and sonographers to facilitate human assessment. However, this work may still remain gaps between real-world clinical utilization due to medical image analysis being a low failure tolerance application. The meaning of this work is to present a new direction for cardiac motion tracking, which is different from the conventional approach. In the current stage, the trained model in public datasets and the results presented are not specific to provide support for clinical use.

## B  Additional Quantitative Results and Visualization

We provide the experiment reported on Hussdorf Distance (HD) for datasets CardiacUDA [17], CAMUS [18] and ACDC [19] in Tables B1, B2 and B3, respectively. Furthermore, the consequent tracking results of 3D echocardiogram and 4D Cardiac MRI are presented in Figures B1 and B2.

Table B1: The performance of different registration methods in CardiacUDA [17]. Results report in Structures (LV, RV, LA, RV) and overall averaged Dice score (Avg. %). Segmentation results are reported in the Hussdorf Distance (HD). **Bold**, underline denote the best results and the second-best performance, respectively.

| Methods | CardiacUDA | | | | |
| (256×256) | LV | RV | LA | RA | Avg. |
| --- | --- | --- | --- | --- | --- |
| | Non-rigid Registration | | | | |
| LDDMM [6] | $18.12_{\pm3.9}$ | $17.33_{\pm3.2}$ | $16.94_{\pm3.4}$ | $16.77_{\pm3.6}$ | $17.27_{\pm3.0}$ |
| RDMM [8] | $17.47_{\pm3.2}$ | $17.69_{\pm3.4}$ | $16.36_{\pm3.2}$ | $15.97_{\pm2.9}$ | $17.01_{\pm2.3}$ |
| ANTs (SyN) [24] | $17.02_{\pm7.5}$ | $16.44_{\pm5.6}$ | $15.81_{\pm4.6}$ | $16.35_{\pm6.1}$ | $16.42_{\pm2.5}$ |
| | Deep Learning Based Registration | | | | |
| VM-SSD [10] | $17.09_{\pm3.3}$ | $16.11_{\pm2.6}$ | $15.60_{\pm2.8}$ | $15.46_{\pm3.2}$ | $16.11_{\pm2.4}$ |
| VM-NCC [10] | $16.84_{\pm2.9}$ | $15.78_{\pm3.1}$ | $15.93_{\pm3.4}$ | $15.16_{\pm2.3}$ | $15.88_{\pm2.5}$ |
| SYMNet [36] | $15.63_{\pm2.7}$ | $15.24_{\pm2.9}$ | $16.18_{\pm3.6}$ | $15.52_{\pm2.4}$ | $15.67_{\pm2.6}$ |
| VM-DIF [9] | $16.16_{\pm3.1}$ | $15.11_{\pm2.7}$ | $16.02_{\pm4.4}$ | $15.68_{\pm3.1}$ | $15.73_{\pm2.3}$ |
| Ahn,S.S, et al. [31] | $16.63_{\pm2.7}$ | $15.13_{\pm3.0}$ | $16.45_{\pm3.1}$ | $14.91_{\pm2.5}$ | $15.48_{\pm2.6}$ |
| DiffuseMorph [12] | $16.26_{\pm2.7}$ | $15.28_{\pm2.5}$ | $14.60_{\pm3.2}$ | $14.77_{\pm3.4}$ | $15.54_{\pm3.2}$ |
| DeepTag [15] | $15.81_{\pm2.8}$ | $15.68_{\pm1.9}$ | $14.39_{\pm2.6}$ | **$13.70_{\pm2.4}$** | $14.94_{\pm2.4}$ |
| GPTrack-M(Ours) | **$14.63_{\pm2.6}$** | **$14.77_{\pm3.0}$** | **$12.19_{\pm2.5}$** | $13.94_{\pm2.1}$ | **$13.87_{\pm2.3}$** |

Table B2: The performance of different registration methods in CAMUS [18] dataset. Results report in Structures (LV, RV, Myo) and overall averaged Dice score (Avg. %). Segmentation results are reported in the Hussdorf Distance (HD). **Bold**, underline denote the best results and the second-best performance, respectively.

| Methods | CAMUS | | | |
| (256×256) | LV | RV | Myo | Avg. |
| --- | --- | --- | --- | --- |
| | Non-rigid Registration | | | |
| LDDMM [6] | $7.210_{\pm3.8}$ | $10.65_{\pm7.7}$ | $6.592_{\pm2.3}$ | $7.305_{\pm2.5}$ |
| RDMM [8] | $6.307_{\pm3.4}$ | $9.584_{\pm6.7}$ | $6.911_{\pm2.5}$ | $6.831_{\pm2.6}$ |
| ANTs (SyN) [24] | $6.644_{\pm3.7}$ | $10.29_{\pm8.3}$ | $6.134_{\pm2.3}$ | $7.166_{\pm2.3}$ |
| | Deep Learning Based Registration | | | |
| VM-SSD [10] | $5.769_{\pm3.0}$ | $8.755_{\pm7.6}$ | $5.231_{\pm1.9}$ | $6.240_{\pm1.8}$ |
| SYMNet [36] | $5.544_{\pm3.5}$ | $9.131_{\pm7.2}$ | $5.466_{\pm2.8}$ | $6.499_{\pm2.0}$ |
| VM-NCC [10] | $5.454_{\pm3.3}$ | $9.094_{\pm7.5}$ | $5.190_{\pm2.1}$ | $6.521_{\pm2.1}$ |
| VM-DIF [9] | $5.382_{\pm2.8}$ | $8.749_{\pm6.6}$ | $5.137_{\pm1.6}$ | $6.304_{\pm1.6}$ |
| Ahn,S.S, et al. [31] | $5.628_{\pm3.2}$ | $8.701_{\pm7.3}$ | $5.478_{\pm1.9}$ | $6.072_{\pm1.7}$ |
| DiffuseMorph [12] | $5.396_{\pm2.7}$ | $8.358_{\pm6.4}$ | $5.066_{\pm1.8}$ | $5.854_{\pm1.8}$ |
| DeepTag [15] | $5.207_{\pm3.1}$ | $7.651_{\pm6.1}$ | **$4.870_{\pm2.2}$** | $5.388_{\pm1.6}$ |
| GPTrack-M(Ours) | **$4.722_{\pm2.8}$** | **$6.857_{\pm5.7}$** | $4.904_{\pm1.8}$ | **$4.945_{\pm1.1}$** |

Table B3: The performance of different registration methods in ACDC [19] dataset. Results report in Structures (LV, RV, Myo) and overall averaged Dice score (Avg. %). Segmentation results are reported in the Hussdorf Distance (HD). **Bold**, underline denote the best results and the second-best performance, respectively.

| Methods (256×256) | ACDC | | | |
|---|---|---|---|---|
| | LV | LA | Myo | Avg. |
| | Non-rigid Registration | | | |
| LDDMM [6] | $5.817_{\pm2.4}$ | $6.662_{\pm2.9}$ | $6.337_{\pm2.8}$ | $6.562_{\pm2.1}$ |
| RDMM [8] | $5.430_{\pm2.5}$ | $6.268_{\pm2.4}$ | $5.953_{\pm2.2}$ | $5.728_{\pm1.5}$ |
| ANTs (SyN) [24] | $5.676_{\pm2.3}$ | $6.547_{\pm2.6}$ | $6.211_{\pm2.7}$ | $6.242_{\pm1.6}$ |
| | Deep Learning Based Registration | | | |
| VM-SSD [10] | $4.708_{\pm1.8}$ | $4.814_{\pm1.7}$ | $5.647_{\pm2.4}$ | $4.942_{\pm1.2}$ |
| VM-NCC [10] | $4.745_{\pm2.1}$ | $5.153_{\pm1.9}$ | $5.231_{\pm2.6}$ | $5.336_{\pm1.3}$ |
| VM-DIF [9] | $4.466_{\pm2.1}$ | $4.782_{\pm1.9}$ | $5.365_{\pm2.6}$ | $4.802_{\pm1.5}$ |
| SYMNet [36] | $4.864_{\pm2.3}$ | $5.149_{\pm2.1}$ | $5.552_{\pm2.7}$ | $5.254_{\pm1.9}$ |
| NICE-Trans [52] | $4.626_{\pm1.9}$ | $4.805_{\pm2.1}$ | $5.096_{\pm2.4}$ | $4.993_{\pm1.6}$ |
| CorrMLP [53] | $3.850_{\pm1.8}$ | $4.061_{\pm1.7}$ | $3.653_{\pm2.4}$ | $3.812_{\pm1.3}$ |
| DiffuseMorph [12] | $4.102_{\pm1.9}$ | $4.054_{\pm2.3}$ | $4.184_{\pm2.0}$ | $3.977_{\pm1.2}$ |
| DeepTag [15, 16] | $3.336_{\pm1.6}$ | $3.651_{\pm2.1}$ | $3.284_{\pm2.2}$ | $3.552_{\pm1.3}$ |
| Transmatch [54] | $3.904_{\pm1.9}$ | $3.855_{\pm2.1}$ | $3.770_{\pm1.9}$ | $3.716_{\pm1.4}$ |
| GPTrack-M(Ours) | $3.285_{\pm1.4}$ | $3.170_{\pm1.8}$ | $3.030_{\pm1.8}$ | $3.361_{\pm1.1}$ |
| FSDiffReg [11] | **$2.970_{\pm1.3}$** | $3.298_{\pm2.0}$ | $2.862_{\pm1.8}$ | $3.283_{\pm1.2}$ |
| GPTrack-XL(Ours) | $3.147_{\pm1.5}$ | **$3.028_{\pm1.9}$** | **$2.844_{\pm1.8}$** | **$3.145_{\pm1.1}$** |

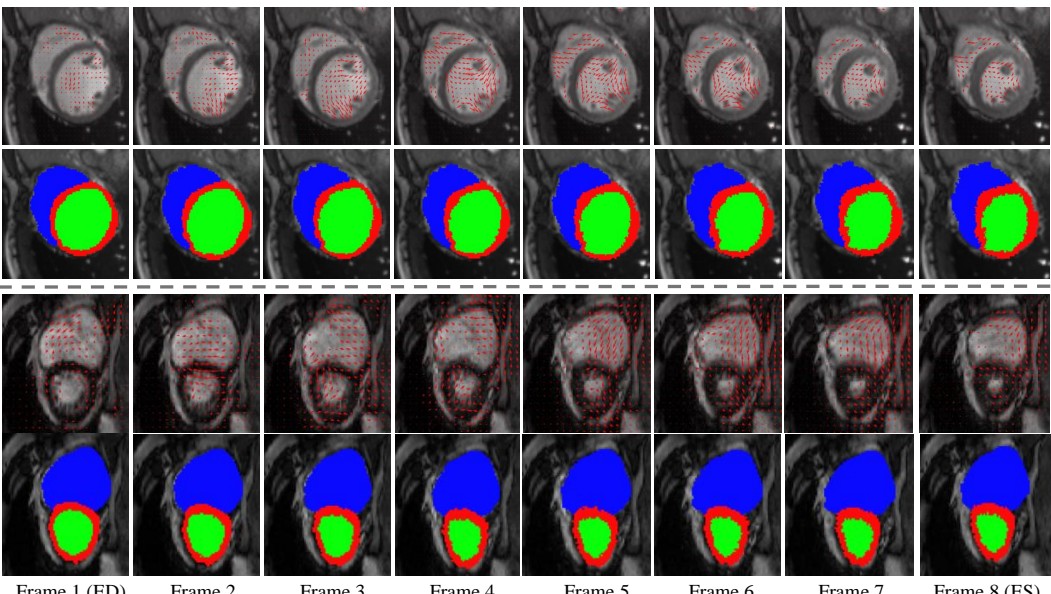

| Frame 1 (ED) | Frame 2 | Frame 3 | Frame 4 | Frame 5 | Frame 6 | Frame 7 | Frame 8 (ES) |

Figure B1: The visualization in 4D Cardiac MRI of estimated motion field and motion tracking results. We visualised the tracking result of the first frame (ED) to the last frame (ES) in ACDC [17]. Colours Red, Blue, and Green denote cardiac structures MYO, LA, and LV, respectively.

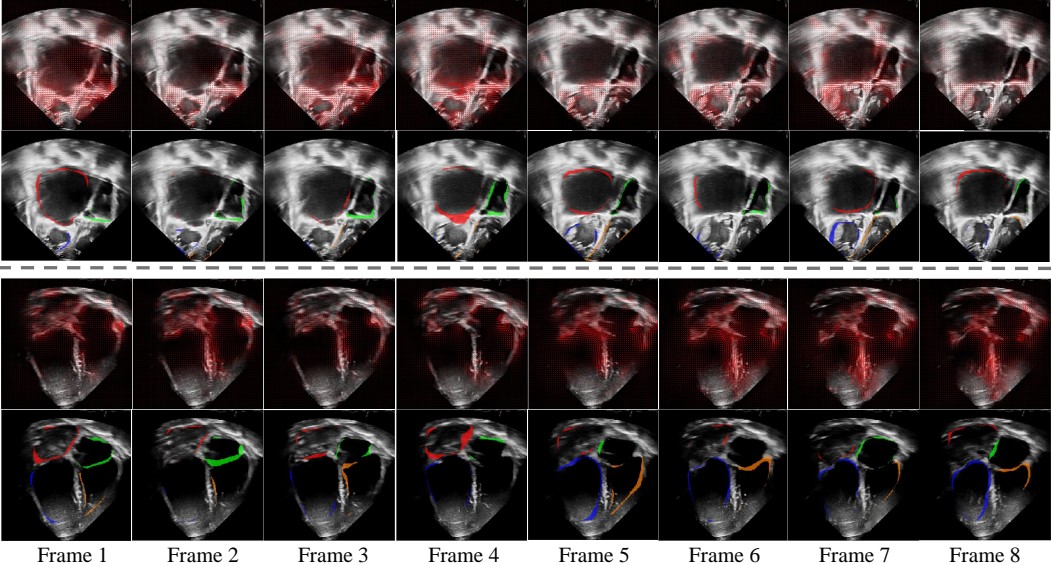

| Frame 1 | Frame 2 | Frame 3 | Frame 4 | Frame 5 | Frame 6 | Frame 7 | Frame 8 |

Figure B2: The visualization in 3D Echocardiogram video of estimated motion field and motion tracking error. We visualised tracking results from the first frame to the last frame, with ground truth from 8 consecutive frames in CardiacUDA [17]. Colours Red, Blue, Green and Orange denote cardiac structures RA, RV, LV and LA, respectively.

