# OpenReview forum: "Bidirectional Recurrence for Cardiac Motion Tracking with Gaussian Process Latent Coding"
_NeurIPS.cc/2024/Conference — NeurIPS 2024 poster_

### Official Review · Reviewer_Tej6 · 2024-07-04

**Soundness:** 3
**Presentation:** 2
**Contribution:** 3
**Rating:** 4
**Confidence:** 4

**Summary:**

To capture the long-term relationship in cardiac motion, the authors proposed GPTrack, a novel unsupervised framework crafted to fully explore the temporal and spatial dynamics of cardiac motion. They proposed employing the sequential Gaussian Process in the latent space and aggregating sequential information in a bidirectional recursive manner. Through experiments on 3D Echocardiogram videos and 4D temporal MRI datasets, they demonstrate that they achieve state-of-the-art performance.

**Strengths:**

1.	employs the Gaussian Process (GP) to promote temporal consistency and regional variability
2.	capture the long-term relationship of cardiac motion via a bidirectional recursive manner

**Weaknesses:**

1. Some details are lacking or not clear in the manuscript, regarding the data processing, how the proposed method is applied to the complete cardiac cycle, and the detailed difference in training and architecture between the proposed method and traditional methods.
2. Lack of comparison to the latest registration methods.
3. No statistical significance analysis is provided.
4. More detailed discussions about the potential application and limitations are needed.

**Questions:**

1.	How is the proposed method applied to the complete cardiac cycle?  What's the difference between the proposed method and the traditional DL-based registration network? Is the comparison to previous pairwise registration approaches a fair comparison?
2.	Does the author use a pair of images or a complete cycle of images as the input? If using the complete cardiac cycle, the incorporation of additional temporal information would significantly increase the computational cost and registration parameters, what is the potential application of the proposed methodology?
3.	The cardiac cycle of different subjects contains different frames, how did the authors process the dataset and train the proposed method?
4.	Why separate the original images into several patches? It would lead to significant unalignment if the motion is large.
5.     Are the improvements significant?

**Limitations:**

The authors only mentioned one sentence of limitation in the conclusion, which needs further discussion.

---

> ### Author Rebuttal · Authors · 2024-08-06
>
> ***Dear reviewer Tej6***: Thank you for your kind comments and suggestions, which help us improve our paper's quality. Here are our responses to weaknesses and questions. New experiments are included in the uploaded PDF file to better illustrate your questions.
>
> &nbsp;
>
> **Q1. Method applied to complete cardiac cycle**
>
> The video frames of the complete cardiac cycle can be continuously fed into our model like the recurrent neural networks (RNNs). The outputs are the deformation between two adjacent frames. Specifically, motion tracking can be regarded as a task of continuous registration and computing the diffeomorphism between every adjacent frame of a video. The motion tracking task is we give the video with $t$ frames ($x_0, x_1, \cdots, x_{t-1}, x_t$) of the complete cardiac cycle, and we output the diffeomorphism ($\phi_1, \cdots, \phi_{t-1}, \phi_t$), where $\phi_i$ indicates the movements from $(i-1)$-th frame to $i$-th frame. In this paper, we strictly follow the previous research [15, 16] and ensure that all the input and output are consistent in experiments.
>
> **Q2. The difference between... Is it a fair comparison?**
>
> As the response in **Q1**, the input of our approach and the traditional DL-based registration network remains the same, i.e., $\phi_1=\text{net}(x_0, x_1), \phi_2=\text{net}(x_1, x_2), \cdots, \phi_t=\text{net}(x_{t-1}, x_t)$.  The traditional DL-based registration network computes the diffeomorphism between only two adjacent frames. In comparison, by employing the recursive manner, our approach can aggregate and maintain the historical temporal information of cardiac motion by hidden state. The introduction of the Gaussian Process can also promote temporal consistency and regional variability in compact latent space. As shown in experimental results, our method reaches higher tracking accuracy while maintaining lightweight inference speed, computational consumption and model parameters. We ensure that the higher performance can be conducted by our approach under the same computational resource compared to those traditional DL-based registration network.
>
> **Q3. A pair of images or a complete cycle of images as the input?**
>
> Please see our posed comment below.
>
> **Q4. Comparison to the latest registration methods.**
>
> We have added three state-of-the-art registration/tracking methods [56, 57, 58] in our experiments on the ACDC dataset. Please see Table R.1.
>
> **Q5. Significantly increases the computational cost and parameters...**
>
> As the response in **Q2**, our experiments illustrate that our approach will not significantly increase computational cost and model parameters. This is because our method can recursively accept the pair-wise frame as input step by step and each input shares the same network as the RNNs. This indicated that our method has a similar inference speed, computational consumption and model parameters as those other methods. All result showcases the significant improvement in tracking and registration accuracy.
>
> **Q6. Potential applications**
>
>  As we discussed in the section *Appendix A2. Broader Impacts*, the cardiac motion tracking can help estimate and quantify myocardial motion within a cardiac cycle. It is a cost-efficient and effective approach for assessing cardiac function. [59] has utilized cardiac motion to efficiently predict human survival. [60] apply motion tracking for physiological motion compensation systems. Motion tracking has also been used for the identification of dilated cardiomyopathy patients in [61]. Our approach can improve the performance of tracking algorithms, and improve the precision of the myocardial motion tracking in model evaluation. Also, we provide motion information to radiologists and sonographers to facilitate human cardiac assessment.
>
> **Q7. How do authors process the dataset and train the proposed method?**
>
> Initially, we select 32 frames from each video, crop and resize all frames to $256\times 256$ in weight and height. For an input video, the input size is $256\times 256\times 32$. During training, the batch size will be set as 1, and our network will take the first frame $x_0$ and initial hidden state $h_0$ (zero vector) as inputs, generating the feature $f_0$ and hidden state $h_1$. For the next time step, the second frame $x_1$ and $h_1$ will be input to the network and generate the $f_2$ and $h_2$. With such a recursive manner, the feature $f_t$ is always generated by $x_{t}$ and $h_{t}$, where the $h_{t}$ help maintain and aggregate the temporal feature information of cardiac motion from $0$ to $t$ moment. The bidirectional process shares the same pipeline as the above description. For the optimization of the training, we use the loss in section 3.5.
>
> **Q8. Why separate the original images into several patches? ...**
>
> 1). Based on our observations of cardiac motion, we first found that cardiac motion has only a small displacement in Echocardiography/MRI images. The position of cardiac main structures remains roughly unchanged throughout images of the same person *(See Appendix Figures B1 and B2)*. 2). We do not simply decompose images into patches, we adapt multiple convolutional layers to convert the images into small patches. Through the receptive field brought by the convolution kernel, we can build the relationship within patches and all patches are not isolated.
>
> **Q9. Are the improvements significant...**
>
> Our method aims to provide a new approach by integrating temporal information via the recursive manner and Gaussian Process to make more accurate velocity field estimation. Compared to other methods, [36] reports a 1%-1.5% increase compared to [10], and [16] demonstrates only 1% improvement compared to the best method. Our method surpasses the SOTA method with 1%-3% accuracy in 3D and 4D datasets, which illustrates our improvements are significant. Furthermore, these improvements may also contribute to other tasks that may related to cardiac function assessment as illustrated in **Q6**.

---

> ### Author Response · Authors · 2024-08-07
> **Thank you for your review and here are some additional information of our rebuttals.**
>
> Thank you for your review and valuable feedback, we will address your concerns in the following. Our new experiments are included in the ***uploaded PDF file*** to better illustrate your questions. All the ***references and citations*** are included in the content of the paper's rebuttal at the beginning.
>
> We thank you very much for all your efforts and valuable time!
>
> **Q3. A pair of images or a complete cycle of images as the input?**
>
> The input of our model is a sequence of images (frames). However, those frames are not inputted all at once, instead, they are inputted one by one like RNNs and our model is shared for different input frames. The number of frames depends on the task requirements. For example, if we want to estimate the motion of the entire cardiac cycle, the input sequence includes the video frames of the complete cardiac cycle. But if we just want to estimate the motion between two frames, it is ok to only input them in order. In our experiments, motion tracking requires the estimation of the entire cardiac cycle that follows previous works [15, 16] settings. Hence, we unsupervised train network with the entire cardiac cycle as [15, 16], and inference the result between every two adjacent frames in each time step.

---

> ### Author Response · Authors · 2024-08-11
> **We have provided a comprehensive response for your question and concerns and we look forward to hearing from you.**
>
> **Dear Reviewer Tej6:**
>
> Thank you again for your valuable comments and your efforts. As the author-reviewer discussion period is coming to a close, we know that there exist some concerns that need to be addressed and clarified. In our rebuttal, we have provided a comprehensive response to your concerns through detailed explanations and thorough experimental validations. May we know if the responses have addressed your concerns?
>
> Thank you for your time and consideration.
>
> Best regards

---

> ### Author Response · Authors · 2024-08-12
> **Your feedback is really important to us**
>
> **Dear Reviewer Tej6:**
>
> We thank you very much that you can review our paper and provide valuable feedback that helps improve our work. As the author-reviewer discussion period is coming to a close, we would like to know whether our responses have addressed your concerns. We know that the discussion may take up your time. However, your feedback is really important to us, and we would like to address your question in our discussion further.
>
> Thank you for your time and consideration, we look forward to hearing from you.
>
> Best regards

---

> ### Author Response · Authors · 2024-08-13
> **The deadline for the author-reviewer discussion period is now approaching, and we are looking forward to hearing from you.**
>
> **Dear Reviewer Tej6:**
>
> The deadline for the author-reviewer discussion period is now approaching. We understand your time is very valuable and we appreciate all your efforts in writing reviews. During the remaining discussion period, we are still waiting for your response and hope our comprehensive response can address your concerns. We will also provide detailed answers to your other questions or concerns.
>
> We all thank you for your time and consideration, and we are looking forward to hearing from you.
>
> Best regards

---

> ### Author Response · Authors · 2024-08-14
> **We are here always ready to answer and address your questions or concerns.**
>
> **Dear Reviewer Tej6:**
>
> The deadline for the author-reviewer discussion period is almost close. Thank you for your efforts and we are all waiting for your responses. We are here always ready to answer and address your questions or concerns.
>
> Thank you for your time and consideration, and we are looking forward to hearing from you.
>
> Best regards

---

### Official Review · Reviewer_bQtH · 2024-07-12

**Soundness:** 3
**Presentation:** 3
**Contribution:** 3
**Rating:** 6
**Confidence:** 4

**Summary:**

1. This paper presents GPTrack, an unsupervised framework designed to thoroughly investigate the temporal and spatial dynamics of cardiac movement.
2. GPTrack refines motion tracking by utilizing sequential Gaussian Processes within the latent space and encoding statistical data with spatial information at each time point, thereby robustly fostering temporal consistency and accommodating spatial variations in cardiac dynamics.
3. This paper aggregates sequential data in a bidirectional, recursive fashion, emulating the principles of diffeomorphic registration to more effectively capture the enduring motion relationships across cardiac areas, including the ventricles and atria.
4. The method enhances the accuracy of motion tracking, all while preserving computational efficiency.

**Strengths:**

1. This paper presents a new framework named GPTrack to improve cardiac motion tracking accuracy.
2. GPTrack innovatively aggregates both forward and backward temporal information by using the bidirectional recurrent transformer.
3. This paper leverages the Gaussian Process to model the variability and predictability of cardiac motion.
4. This paper demonstrates the state-of-the-art in 3D echocardiogram and 4D cardiac MRI datasets.
5. The methodology employed in this study is robust and rigorous.
6.The implications of the findings are clearly articulated and show practical relevance.

**Weaknesses:**

1. Line 26, Can optical flow not capture temporal coherence? I doubt that's the case.
2. Line 31, The citation is not formatted correctly.
3. Line 61, "contribuion" repeats the idea of "facets" mentioned earlier, which is redundant.
4. There is no experimental comparison with methods such as denoising diffusion probabilistic models (DDPM).
5. Line 111, How is the relationship between the diffeomorphism \phi_t and the method proposed in the paper demonstrated?
6. The experiment does not include a comparison with the method in reference [36].
7. The paper does not explain, demonstrate, or theoretically analyze long-term dependencies, nor does it clarify the duration of "long-term."
8. Many acronyms are repeated unnecessarily.
9. Line 135, What is meant by "motion consistency," and how do you define and assess its consistency?
10. Line 148, Here, x_{t} represents features; it should not use the same mathematical symbol as the raw data to avoid confusion.
11. Figure 3, in the right subplot, what is meant by "elu(x)"?
12. Line 206, the formula below this line is not numbered. But the author refers to this equation.
13. How is the multidimensional z_{t} output generated? Multidimensional z_{t} involves multitask Gaussian processes, which are not mentioned in this paper.
14. Line 221, sometimes \mu_{t} is a scalar, and sometimes it is a vector.
15. Line 258 and Line 272, there is a conflict. Why is this done? Please explain the motivation.
16. Figure 6, Why does the error decrease and then increase? Moreover, this trend is similar even across different methods.
17. There should be a comparison with optical flow-based methods.
18. Why is FSDiffReg only included in Table B3?
19. The discussion of results is somewhat superficial. A deeper analysis linking the findings back to the research questions and theoretical framework would enrich the paper's contribution to the field.
20. There are inconsistencies in the terminology used throughout the paper. Consistently using the same terms would improve clarity and reduce confusion for the reader.
21. The paper would benefit from a stronger connection between the theoretical framework and the empirical results.

**Questions:**

Included in the Strengths and Weaknesses.

**Limitations:**

Included in the Strengths and Weaknesses.

---

> ### Author Rebuttal · Authors · 2024-08-06
>
> ***Dear Reviewer bQtH:***
> Thank you for your valuable feedback, we would like to address your questions point by point in the following.
>
> &nbsp;
>
> **Q1. Line 26, Can optical flow ...**
>
> The optical flow (OF) is also able to capture the temporal coherence. In *section Appendix A1*, we discussed the difference between OF-based and Diffeomorphism Mapping methods. We consider that OF-based methods do not necessarily preserve topology, non-globally one-to-one (objective) smooth and continuous mapping with invertible derivatives. These attributes do not match the description of human cardiac which is considered as an incompressible material. We also add a new experiment compared to optical flow-based methods in Table R.3 of our rebuttal pdf.
>
> **Q4. No experimental comparison with DDPM ...**
>
> In our experiments, we compared two state-of-the-art DDPM methods [11, 12]. As shown in Tables R.1 and R.2. Our GPTrack can reach more accurate results compared to DDPM methods with faster inference speed and less computational consumption.
>
> **Q5. The relationship between the $\phi_t$...**
>
> The diffeomorphism $\phi_t$ is the $t$-th tracking output of our method, which is differentiable and invertible mappings generated by the $t$-th and ($t$-1)-th frames.
>
> **Q6. Comparison with method [36]...**
>
> We have added the corresponding experiment in Tables R.1 and R.2.
>
> **Q7 . Clarify of "long-term."...**
>
> As illustrated in Figure 2 of our manuscript, the "long-term" indicates that the historical information of the cardiac motion will be maintained and encoded as features by our framework. Previous conventional methods only consider the relationship between two adjacent frames, while ours can aggregate the features across long intervals bidirectionally.
>
> **Q9. Line 135, the meaning of "motion consistency", define and assess consistency**
>
> Sorry that our description of "motion consistency" makes you confused. Motion consistency is the consistency of movements between two adjacent state spaces across different individuals. The state space is the set of all possible states of the human cardiac motion. We consider that cardiac motion usually has a certain principle. For example, as shown in Figure 2 in our manuscript, the Diastole and Systole always occur in each heartbeat cycle of every person. The movement of each organ (Ventricle, Atrium, Myocardium, \emph{et al.}) always has its fixed direction, speed and range. Those motions can be considered as "consistency" and formulated as temporal patterns for learning. In our approach, we utilize the recursive manner to aggregate the long-term temporal feature without increasing the time complexity. Also, we employ the Gaussian Process to capture the temporal correlation and heterogeneity in cardiac motion, and provide predictions and interpolation. These designs help promote temporal consistency and regional variability in compact latent space.
>
> **Q11. Figure 3, what is meant by "elu(x)"...**
>
> As described in line 166, the $elu(\cdot)$ is the exponential linear unit [42]. We will make it clearer in our final version.
>
> **Q12. Line 206, the formula below this line is not numbered...**
>
> Thanks for your kind reminder. We have corrected this typo.
>
> **Q13. How is $z_{t}$ generated ...**
>
> Firstly, as mentioned in Line 203, we assign independent GP priors to all values in the latent coding of $\mathbb{R}^{P \times C}$  dimension, indicating $R \times C$ scalar Gaussian processes are independently assigned across $T$ time stamps. Due to our independent modelling, we take one scalar Gaussian process $\\{z_t\\} _{t=1}^T$  for illustration, whose corresponding observation is $\\{f_t\\} _{t=1}^T$. Then for each scalar sequential Gaussian process, within the framework of representing sequential Gaussian process with Kalman filtering, we use the initial condition $\textbf{z}_0$ calculated in Line 220 and mean-variance calculated by Equation 10 to update $\textbf{z}_t := \boldsymbol{\mu}_t,~t=1,...,T$. To sum up, the multidimensional $z_t$ comes from aggregating totally $R \times C$ scalar Gaussian process together due to our independent modelling, and each individual is sequentially updated by the Kalman filtering of Equation 10.
>
> **Q14. Line 221, sometimes $\mu_{t}$ is a scalar, and sometimes it is a vector.**
>
> Thanks for your kind reminder. Here $\boldsymbol{\mu}_{t}$ together with Equation 10 below are all vectors, and we have corrected this typo.
>
> **Q15. Conflict in Line 258 and Line 272...**
>
> The $368 \times 368$ in Line 272 is a typo, which should be the same size $384 \times 384$ as Line 258. We will correct them in the final version.
>
> **Q16. The error decreases and then increases in Figure 6...**
>
> In Figure 6, the error is calculated between the $t$-th tracking result tracked from the initial (reference) frame and the ground truth of the $t$-th frame. Since the human heartbeat is a reciprocating motion (refer to Figure 1), while the heartbeat state of the current frame is close to the initial frame, the registration between these two frames is easier, and therefore the smaller the error (the higher the Dice value).
>
> **Q17. Comparison with OF-based methods...**
>
> Please see the Table R.3 for more details.
>
> **Q18.  FSDiffReg is only included in Table B3.**
>
> Sorry that we only report FSDiffReg in Table B3 and Table 3 since FSDiffReg [11] only provides a 3D network for MRI images and reports the experiment in 4D MRI images. As shown in Table R2, we further the FSDiffReg in 3D echocardiogram video and report its motion tracking results.
>
> **Q2,3,8,10,19,20 and 21. Typos / Citation not formatted correctly / Redundant word / Acronyms repeated unnecessarily / Superficial discussion of results /  Inconsistencies terminology and structures.**
>
> Thank you very much for pointing out the drawbacks and helping further improve our work. We will carefully check the paper to avoid typos and reorganize the connection and details to make it more coherent and easy to read.

---

> ### Author Response · Authors · 2024-08-07
> **Thank you for your review and here are some additional information of our rebuttals.**
>
> We appreciate your review and valuable feedback, we will address your concerns in the rebuttal. Our new experiments are included in the ***uploaded PDF file*** to better illustrate your questions. All the ***references and citations*** are included in the content of the paper's rebuttal at the beginning.
>
> Thank you for your efforts and valuable time!

---

> ### Comment · Reviewer_bQtH · 2024-08-08
>
> The rebuttal addressed my concerns

---

> > ### Author Response · Authors · 2024-08-08
> >
> > Thank you very much again for your time and efforts, we are delighted that our rebuttal has addressed your concerns! Your suggestions and questions help us better improve our work. We believe that our research can further contribute to cardiac motion tracking. We expect that the employment of the recursive manner as well as the Gaussian Process can bring more insight to the follow-up research related to medical image analysis.

---

### Official Review · Reviewer_ssxr · 2024-07-14

**Soundness:** 3
**Presentation:** 3
**Contribution:** 3
**Rating:** 7
**Confidence:** 5

**Summary:**

The authors proposed a latent modeling framework for cardiac motion tracking. They introduced the GPTrack module for image encoding, which considers both forward and backward information flow. A Gaussian process was integrated to describe the motion prior. Extensive experiments were performed on both echocardiography and cardiac MRI datasets, demonstrating good results.

**Strengths:**

1.A transformer-based image encoder that integrates bidirectional information flow, considering long-range relationships.
2.A Gaussian process latent modeling approach to describe the dynamics of cardiac motion.
3.Competitive results on both echocardiography and cardiac MRI (CMR) datasets.

**Weaknesses:**

1.It appears that the authors misunderstood the cardiac MRI in Figure 2. The short-axis CINE image shows the myocardium of the left ventricle (green) and right ventricle (red) rather than the left atrium.
2.The authors used PSNR, SSIM, and DICE to evaluate motion tracking performance, which is reasonable. However, they did not evaluate the physics/physiological plausibility. For the myocardium, which is considered an incompressible material, the determinant of the Jacobian |J| should be around 1. I suggest the authors add some evaluation comparing the physiological plausibility [1].
[1] doi: 10.1016/j.media.2022.102682

**Questions:**

1.Are there any results showing the impact of varying the number of frames?
2.The myocardium is the tissue of interest for motion tracking, as the blood pool of the left/right ventricle involves flow filling. Could there be more focus on comparing the myocardium?

**Limitations:**

The authors described the limitations appropriately.

---

> ### Author Rebuttal · Authors · 2024-08-07
>
> ***Dear reviewer ssxr:***  Thank you for your efforts and valuable feedback, we would like to address your questions point by point in the following.
>
> &nbsp;
>
> **Q1.Are there any results showing the impact of varying the number of frames?**
>
> Table R.4 shows the ablation study of varying frame numbers on the CardiacUDA dataset. Methods that do not consider the temporal information such as VM-DIF [9] and FSDiffReg [11] would not be affected by the frame length significantly. The DeepTag employs the Lagrangian displacements affected by the frame length, whose performance decreases when reducing the number of frames. Our GPTrack also sees degradation when reducing the number of frames, especially when the input only contains two frames, which leads the motion-tracking task to a pair-wise registration task. However, with the contribution of our designed framework and Gaussian Process, the result of GPTrack can also outperform other methods.
>
> **Q2. The myocardium is the tissue of interest for motion tracking, as the blood pool of the left/right ventricle involves flow filling. Could there be more focus on comparing the myocardium?**
>
> Thank you for your kind suggestion, we highlight the myocardium in Tables R.1 and R.2, in our final version and following research, we will also emphasize the importance of myocardial in cardiac motion tracking.
>
> **Q3. The authors add some evaluation comparing the physiological plausibility.**
>
> Thank you for your helpful suggestions on how to strengthen our paper. We have reported the related metric in Tables R.1 and R.2 and highlighted the results with blue colour (Please see Rebuttal pdf). Related research mentioned by the reviewer [54, 55] for motion tracking with biomechanics-informed prior uses supervised learning and our method employs unsupervised learning for motion-tracking tasks. Hence, in our rebuttal, we may not be able to compare these two works in our experiments.  However, we also consider that physiological plausibility is an important index that can help evaluate the LV myocardial strain estimates and the learnt biomechanical properties. Following the research [54, 55], we compute the mean absolute difference between the Jacobian determinant and 1 ($||J|-1|$) over the tracking areas. In our final version, we will also add these metrics and cite related works in our final revision.
>
> **Q4. It appears that the authors misunderstood the cardiac MRI in Figure 2. The short-axis CINE image shows the myocardium of the left ventricle (green) and right ventricle (red) rather than the left atrium.**
>
> Thank you for pointing out our mistake and misunderstanding, we will be really carefully revising all Figures, Tables and Descriptions. Make sure all content in our paper is presented correctly in our final version.

---

> > ### Comment · Reviewer_ssxr · 2024-08-11
> >
> > Thank you for your rebuttal. My concerns have been addressed. It is a nice paper to read. I have increased my rating to accept.

---

> > > ### Author Response · Authors · 2024-08-11
> > >
> > > We greatly appreciate your efforts and help us improve our work! In our future work, we will further devote ourselves to the cardiac motion tracking task as it is important in cardiac function assessment. Furthermore,  we will follow the previous related research to evaluate the physics/physiological plausibility and make our research and results more reliable in real scenarios. We believe that our work can contribute to follow-up research in medical image analysis.

---

> ### Author Response · Authors · 2024-08-07
> **Thank you for your review and here are some additional information of our rebuttals.**
>
> Thank you very much for your review and valuable feedback. Your suggestions and comments are important to us and we will address all your concerns in the rebuttal. Our new experiments are included in the ***uploaded PDF file*** to better illustrate your questions. All the ***references and citations*** are included in the content of the paper's rebuttal at the beginning.
>
> Thank you for your efforts and valuable time!

---

> ### Author Response · Authors · 2024-08-11
> **May we kindly know if our detailed explanations and more experimental validations have addressed your concerns?**
>
> **Dear Reviewer ssxr:**
>
> Thank you again for your valuable comments and your efforts, your kind suggestions help further our paper's quality. We know that there may still be some concerns that need to be addressed and clarified. In our rebuttal, we have provided a comprehensive response to your concerns through detailed explanations and more experimental validations. As the author-reviewer discussion period is about to close, may we kindly know if the responses have addressed your concerns?
>
> Thank you for your time and consideration.
>
> Best regards

---

### Author Rebuttal · Authors · 2024-08-07

### **We first thank all reviewers for their valuable feedback to help us improve our work**.
### **Below, we will address the concerns of reviewers about the experiments and details of our proposed method. In our uploaded one-page PDF, we provide more experiments and ablation studies to better illustrate the questions.**
### **For point-to-point question answering, we left our rebuttal in the corresponding position for each reviewer.** *All Tables for rebuttal are included in our uploaded one-page PDF.*

&nbsp;
### References:

[54] Qin, C., Wang, S., Chen, C., Qiu, H., Bai, W. and Rueckert, D., 2020. Biomechanics-informed neural networks for myocardial motion tracking in MRI. In Medical Image Computing and Computer Assisted Intervention–MICCAI 2020: 23rd International Conference, Lima, Peru, October 4–8, 2020, Proceedings, Part III 23 (pp. 296-306). Springer International Publishing.

[55] Qin, C., Wang, S., Chen, C., Bai, W. and Rueckert, D., 2023. Generative myocardial motion tracking via latent space exploration with biomechanics-informed prior. Medical Image Analysis, 83, p.102682.

[56] Mingyuan Meng, Dagan Feng, Lei Bi, and Jinman Kim, "Correlation-aware Coarse-to-fine MLPs for Deformable Medical Image Registration," IEEE/CVF Conference on Computer Vision and Pattern Recognition (CVPR), pp. 9645-9654, 2024.

[57] Chen, Z., Zheng, Y. and Gee, J.C., 2023. Transmatch: A transformer-based multilevel dual-stream feature matching network for unsupervised deformable image registration. IEEE transactions on medical imaging, 43(1), pp.15-27.

[58] Meng, M., Bi, L., Fulham, M., Feng, D. and Kim, J., 2023, October. Non-iterative coarse-to-fine transformer networks for joint affine and deformable image registration. In International Conference on Medical Image Computing and Computer-Assisted Intervention (pp. 750-760). Cham: Springer Nature Switzerland.

[59] Bello, G.A., Dawes, T.J., Duan, J., Biffi, C., De Marvao, A., Howard, L.S., Gibbs, J.S.R., Wilkins, M.R., Cook, S.A., Rueckert, D. and O’regan, D.P., 2019. Deep-learning cardiac motion analysis for human survival prediction. Nature machine intelligence, 1(2), pp.95-104.

[60] Richa, R., Poignet, P. and Liu, C., 2008. Efficient 3D tracking for motion compensation in beating heart surgery. In Medical Image Computing and Computer-Assisted Intervention–MICCAI 2008: 11th International Conference, New York, NY, USA, September 6-10, 2008, Proceedings, Part II 11 (pp. 684-691). Springer Berlin Heidelberg.

[61] Puyol-Antón, E., Ruijsink, B., Gerber, B., Amzulescu, M.S., Langet, H., De Craene, M., Schnabel, J.A., Piro, P. and King, A.P., 2018. Regional multi-view learning for cardiac motion analysis: Application to identification of dilated cardiomyopathy patients. IEEE Transactions on Biomedical Engineering, 66(4), pp.956-966.

[62] O'Briain, T., Uribe, C., Yi, K.M., Teuwen, J., Sechopoulos, I. and Bazalova-Carter, M., 2022. FlowNet-PET: unsupervised learning to perform respiratory motion correction in PET imaging. arXiv preprint arXiv:2205.14147.

[63] S. Mocanu, A. Moody, and A. Khademi, “FlowReg: Fast Deformable Unsupervised Medical Image Registration using Optical Flow,”
Machine Learning for Biomedical Imaging, pp. 1–40, Sep. 2021.

---

### Decision · Program_Chairs · 2024-09-25

**Decision:**

Accept (poster)

**Comment:**

This work proposes an unsupervised framework for cardiac motion tracking by considering the temporal and spatial dynamics of cardiac motion. Reviewers recognise that this work has merits in the innovation of model design, the state-of-the-art results, robust and rigorous methodology, and the clear articulation of findings. At the same time, the reviewers raise issues related to more evaluation and comparison, the impact of the number of frames, deeper analysis and stronger connection, and the statistical significant analysis. The authors have made efforts to provide responses to the raised issues. Two of the three reviewers indicate that the rebuttal has addressed their concerns and give positive final ratings. The third reviewer does not respond to the rebuttal and the rating is at borderline reject. The AC agrees that this work has its novelty and values and that the authors have well responded to the reviews. Meanwhile, it is recommended that this work apply statistical significant test (say, paired samples t test) to the results in Tables 2 and 3 to better assess the improvement. Considering all the factors, AC recommends this paper for acceptance. The authors shall effectively incorporate the key information in the rebuttal to further strengthen this work.